# SparseInfer: Accelerating Large Language Model Inference with Semantics-Inspired Adaptive Sparse Activation

Qinsi Wang [1]  Saeed Vahidian [1]  Hancheng Ye [1]  Jianyang Gu [2]  Jianyi Zhang [1]  Yiran Chen [1]

## Abstract

Large Language Models (LLMs) with billions of parameters have transformed AI applications but require immense computational and memory resources during inference. Adaptive sparse activation inference, which activates only a small number of neurons for each token, offers a novel way to accelerate model inference without degrading performance, showing great potential for resource-constrained hardware devices. Yet existing token-level MLP-based methods frequently alter activation maps, reducing efficiency gains. In this paper, we introduce **SparseInfer**, an MLP-free adaptive sparse activation inference method based on sentence-level prediction. We first propose the concept of core neurons and empirically demonstrate that, for an input sentence, LLMs only need the core neurons to maintain performance. Remarkably, we discovered that core neurons exhibit both stability and similarity in relation to the sentence's semantics—an insight overlooked by previous studies. Building on this finding, we design two semantic-based methods for predicting core neurons to fit different input scenarios, which enables core neurons to be determined during the pre-filling stage and fixed during the encoding stage. Our experiments verify SparseInfer exhibits good performance in various tasks and achieve $10.33\times$ speed up. The code is in https://wangqinsi1.github.io/coreinfer_page/

## 1. Introduction

Generative Large Language Models (LLMs) have garnered significant attention for their exceptional abilities in creative writing, advanced code generation, and complex natural language processing tasks (Brown, 2020; Chowdhery et al., 2023; Touvron et al., 2023a; Team et al., 2023). These models have profoundly impacted our daily lives and work practices. A generation task typically involves multiple inferences—a single inference during the pre-filling stage and multiple inferences during the decoding stage—but due to the vast number of parameters in LLMs, executing these inferences becomes highly expensive (Pope et al., 2023). To make generative LLMs more accessible, an increasing number of researchers are focusing on accelerating the inference process. The key challenge is: **how can we optimize both memory usage and computational efficiency for model inference while maintaining performance?**

Model compression (Buciluǎ et al., 2006; Cheng et al., 2017; Choudhary et al., 2020) has been extensively studied to address this issue by transforming the original model into a light version. Representatively, quantization (Lin et al., 2024; Frantar et al., 2022; Dettmers et al., 2024) uses fewer bits to represent parameters, reducing the memory needed for model storage and inference. Pruning (LeCun et al., 1989; Lee et al., 2018; Frankle & Carbin, 2018; Bansal et al., 2022) decreases the computational load during inference by removing unimportant neurons or structural blocks from the model. However, these methods usually break the original structure and trade-off the performance for efficiency. Additionally, due to the diversity of modern hardware, these methods cannot achieve hardware generalization. For instance, although 3-bit quantization has shown potential, most current hardware devices do not support it yet (Cheng et al., 2017; Kim et al., 2021).

Dynamic activation sparse inference (Liu et al., 2023) is another way to accelerate inference without the limitations of model compression. This approach is based on the observation that activations of individual tokens in large language models are often highly sparse (Song et al., 2023). During the decoding stage, dynamic activation sparse inference activates only a small number of neurons for each token, effectively accelerating model inference. This method has already demonstrated significant potential on resource-constrained devices. For instance, PowerInfer (Song et al., 2023) accelerates LLMs inference by $11.6\times$ on PCs by implementing activation prediction and dynamic sparse inference. PowerInfer2 (Xue et al., 2024) and LLM in the

---

[1]Duke University [2]Ohio State University. Correspondence to: Qinsi Wang <qinsi.wang@duke.edu>.

*Proceedings of the $43^{rd}$ International Conference on Machine Learning*, Seoul, South Korea. PMLR 306, 2026. Copyright 2026 by the author(s).

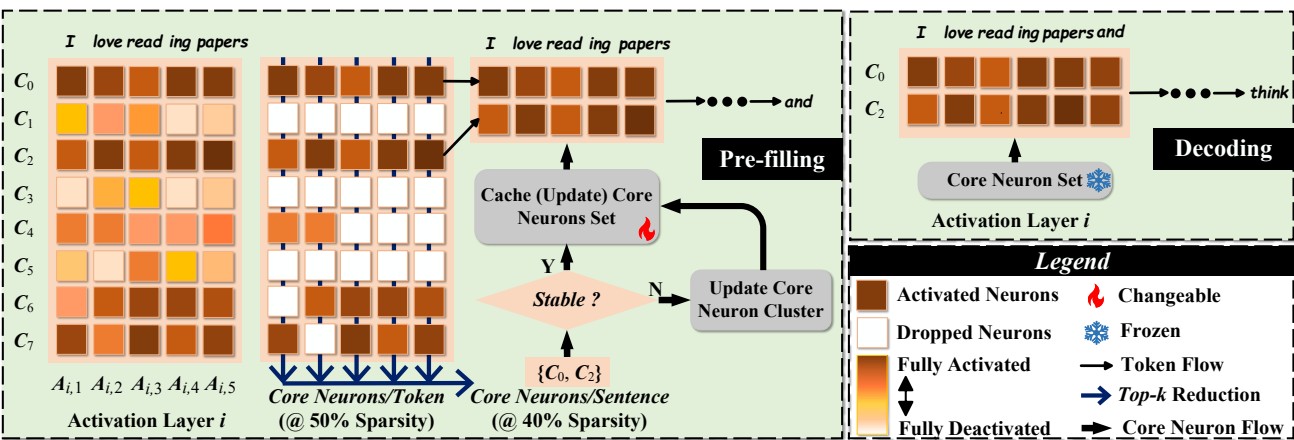

*Figure 1.* The overview framework of SparseInfer. In the pre-filling stage, at each activation layer, taking the $i$-th activation layer as an example, we first extract the token-wise core neurons based on the top-k selection and then further extract the top-k commonly activated core neurons among all tokens, which go through the stability estimation to determine how to update the sentence-wise core neuron set. After determination, the core neuron set will be fixed and utilized for sparse decoding.

Flash (Alizadeh et al., 2023) apply this technique to mobile phones to accelerate LLMs inference on mobile platforms. These methods usually train an MLP predictor in each activation layer to predict neurons that will be activated (Liu et al., 2023; Song et al., 2023; Xue et al., 2024; Alizadeh et al., 2023). Such strategies present two weaknesses: (1) **Irregular and frequent resource calls during decoding** due to the token-wise activation prediction, which may hinder further acceleration of the decoding stage. (2) **Additional computation costs during decoding** due to the introduction of MLP per activation layer, which sometimes cannot be ignored. For example, MLPs will introduce an additional 10% computation cost when applied (Alizadeh et al., 2023).

To this end, aiming at solving the above two problems, we propose **SparseInfer, a novel sparse inference strategy featuring the sentence-wise activation sparsity without additional MLP predictors.** Specifically, we first define a set of core neurons for each sentence, representing the most essential neurons an LLM needs to process it. These core neurons are empirically demonstrated sufficient enough for an LLM to perform lossless tasks. Then, to predict a sentence's core neurons, we explore the relationship between a sentence's core neurons and its semantics. We performed explorations at the level of stability and similarity between core neurons and semantics and found strong correlations in both aspects. Inspired by this, we propose two methods to predict a sentence's core neurons based on its semantics.

Fig. 1 shows our overview and algorithm flow. Notably, for each sentence, SparseInfer only needs to predict the core neurons during the pre-filling stage. During the decoding stage, it consistently uses this set of neurons without needing to repeatedly predict and change the activation map as previous methods do. Moreover, SparseInfer does not use additional MLP predictors, thereby maximizing the potential of sparse inference. In summary, our contributions are as follows:

- We propose SparseInfer, an sentence-level adaptive sparse inference framework, in which we define sentence-wise core neurons as the most essential group of neurons for decoding.

- By exploring the relationship between core neurons and semantics, we discover that core neurons exhibit both stability and similarity in relation to the semantics.

- Through experiments, we demonstrate that our method possesses both model generalization and task generalization. Without degrading performance, it achieves a $10\times$ and $3\times$ acceleration compared to Huggingface and PowerInfer on NVIDIA GPUs, respectively.

## 2. Definition and Exploration of Core neurons

In this section, we first present the definition of core neurons and prove their effectiveness (Sec. 2.1). Then, several exciting insights are observed about the correlation between sentence-wise core neurons and its semantics in both stability and similarity (Sec. 2.2).

### 2.1. Definition and Role of Core Neurons

Motivated by previous works (Alizadeh et al., 2023) attempting to predict the most important neurons for inference and the fact that large activation values in LLMs contribute more to model performance than small ones, we first define token-wise core neurons and extend it to sentence-wise.

**Definition 1: Token-wise Core Neurons.** For a single token $x$ at the $i$-th activation layer of the LLM, the input is denoted as $x_i$. And the activation can be denoted by the vector $A_i(x_i) = [a_1, a_2, \ldots, a_N]$, where $N$ is the number of neurons and $a_n$ is the activation value of the $n$-th neuron. We define the core neurons of $x_i$ as the top $\alpha$ of neurons with the largest positive activation values (i.e., $a_n > 0$).

The core neurons for token $x$ at the $i$-th layer is defined as the top $\alpha$ largest activated neurons, whose set can be

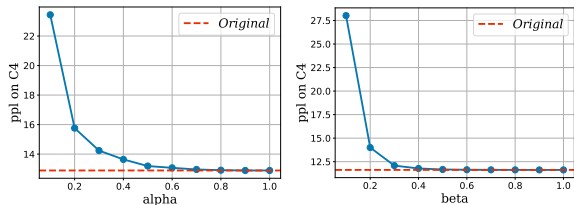

*Figure 2.* The impact of $\alpha$ (left) and $\beta$ (right) on final performance. The experiment is conducted on the OPT 6.7b and C4.

formulated as follows.

$$\mathcal{C}_\alpha(x_i) = \{n \mid a_n \geq \text{Percentile}(A_i^+, \alpha)\}, \quad (1)$$

where $A_i^+ = \{a_n \mid a_n > 0, a_n \in A_i\}$ represents the set of positively-activated neurons at the $i$-th activation layer, and $\text{Percentile}(A_i^+, \alpha)$ denotes the $\alpha$-th percentile of the positive activation.

**Definition 2: Sentence-wise Core Neurons.** For a sentence $s$ containing $M$ tokens, the input of the $i$-th layer is $s_i = [x_i^1, x_i^2, \ldots, x_i^M]$. Based on Equation 1, each $x_i^m$ has core neurons $\mathcal{C}_\alpha(x_i^m)$. We define the core neurons for $s_i$, $\mathcal{C}_\alpha^\beta(s_i)$, as the top $\beta$ of neurons that appear most frequently in the core neurons of all tokens, i.e., $\{\mathcal{C}_\alpha(x_i^1), \mathcal{C}_\alpha(x_i^2), \ldots, \mathcal{C}_\alpha(x_i^M)\}$, thus can be formulated as

$$\mathcal{C}_\alpha^\beta(s_i) = \{n \mid f_\alpha(n; s_i) \geq \text{Percentile}(f_\alpha(s_i), \beta)\}, \quad (2)$$

where $f_\alpha(s_i)$ denotes the count set of each neuron across all tokens, which is formulated as follows.

$$f_\alpha(s_i) = \{f_\alpha(n; s_i)\}_n = \{\sum_{m=1}^{M} \mathbb{I}(n \in \mathcal{C}_\alpha(x_i^m))\}_n, \quad (3)$$

where $\mathbb{I}(\cdot)$ is an indicator function that returns `one` if $n$ is in $\mathcal{C}_\alpha(x_i^m)$ else `zero`. $\text{Percentile}(f_\alpha(s_i), \beta)$ denotes the $\beta$-th percentile of $f_\alpha(s_i)$.

**Effectiveness of Core Neurons.** We test the effectiveness of the proposed core neurons at two levels by experimenting on the C4 (Sakaguchi et al., 2021) with multiple hyperparameter settings. The results are shown in Fig. 2 (a) and (b). As can be seen from Fig. 2 (a), it is exciting that when $\alpha$ and $\beta$ are very low, the model has only a small performance loss. For example, ppl only increases by 2% when $\alpha$ is 0.4. And when $\beta = 0.25$, ppl only increases by 3%.

To understand why the sentence-wise core neurons are effective, we further explore the distribution of token-wise core neurons in different sentences, and the results are shown in Fig. 3 (a). It can be seen that the distribution of core neurons of tokens in the same sentence is always closer (meaning that there are more identical neurons in their core neurons), while the distribution of core neurons of tokens in different sentences shows a clustering phenomenon. This explains why the sentence-wise core neurons are effective: since tokens in the same sentence tend to activate similar

neurons, a small number of core neurons can meet the needs of the entire sentence inference.

This result reveals a powerful potential of core neurons: **For an input sentence, LLMs only need the core neurons to maintain performance.** Different from prior works exploring token-wise sparsity in activation, our work is the first to explore the sentence-wise sparsity in activation.

**2.2. Exploration of Core Neurons**

In the previous section, we defined core neurons and explained their effectiveness. To better predict core neurons, in this section, we explore the relationship between core neurons and the input sentence.

Semantics is a crucial aspect of the information conveyed by the input sentence. Recent studies (Wang et al., 2024) have demonstrated that the similarity of LLMs activation shows a strong correlation with semantic similarity. This prompts us to speculate and explore: Are core neurons related to the semantics of the input sentence? Here we introduce two of our insights into the relationship between semantic and core neurons, respectively related to stability and similarity.

**Insight-1: The Stability of Core Neurons Is Related to Semantic Stability.**

First, we explore the relationship between the stability of core neurons and semantics. To investigate this, we extended sentences of varying lengths with coherent and fluent continuations, subsequently measuring the semantic similarity and core neuron similarity between the original and the extended sentences. The results, illustrated in Fig. 3 (b)(c), reveal a robust correlation between the changes in semantic similarity and core neuron similarity. Notably, when there is a high semantic similarity between an original sentence and its extension, the core neuron similarity is also elevated.

As shown in Fig. 3 (b)(c), we can find that adding 8-token and 64-token continuations to a sentence of 256 tokens does not change the semantics at all (semantic similarity is 1). In this case, the core neurons change by only 3% and 6%, respectively. Furthermore, in Fig. 3 (d), we show the changes in $\mathcal{C}_\alpha^\beta(s_i)$ as the length of a fluent and continuous sentence increases. It can be seen that as the sentence length increases and the semantics become clearer, the core neurons gradually stabilize. Adding more to the sentence at this point does not cause significant changes in the core neurons. In Fig. 3 lower, we visualize the core neurons of the same sentence at different lengths. We can see that core neurons are still changing when the sentence length is less than 100, and when the sentence length is 200 and 300, the core neurons have basically remained unchanged. Thus, our experimental analysis reveals that during the generation process, core neurons tend to remain stable when the semantics of the sentence is consistent.

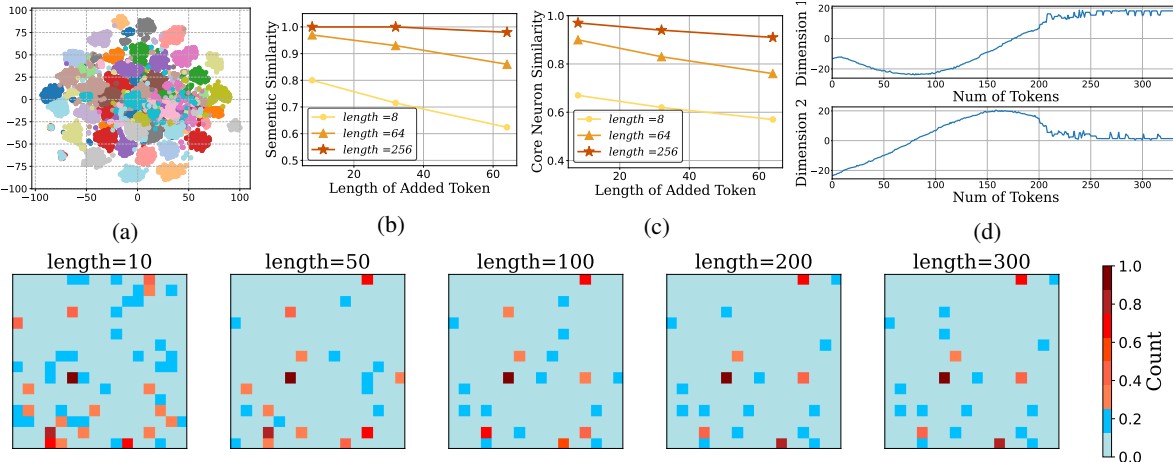

*Figure 3.* (Upper) (a)Clustering of token-wise core neurons in different sentences. We randomly selected 20 sentences from the C4 dataset and observed the activation pattern of the 25-th layer of the model. Each point represents a $\mathcal{C}_\alpha(x_i)$. The same color represents the same sentence. We used t-SNE (Van der Maaten & Hinton, 2008) to reduce the data dimension. (b) (c): When adding tokens after the original sentence, The semantics similarity and core neurons similarity between the extended and the original sentence. (d) Schematic diagram of the change of core neurons as the length of the sentence increases. We use t-SNE to reduce the dimension of core neurons to two dimensions and observe the changes in the dimension 1 and deimension 2. (Lower) Visualization of core neurons when the token length of the continuous input sentence is 10, 50, 100, 200, and 300. We randomly selected 256 neurons in the 25-th layer of the OPT-6.7b model. Each pixel represents a neuron, and the color indicates the frequency of the neuron in all the current $\mathcal{C}_\alpha(x_i)$. $\mathcal{C}_\alpha^\beta(s_i)$ is a part of the neurons with the highest frequency (brightest).

## Insight-2: The Similarity of Core Neuron Is Related to Semantic Similarity.

Furthermore, we investigate the relationship between core neuron similarity and semantic similarity. To illustrate this intuitively, we select the ag_news dataset (Zhang et al., 2015), which contains sentences from four different topics, sentences within the same topic often have closer semantics. We input different sentences from ag_news into the model and observed the distribution of their core neurons. Semantic similarity is measured by using Sentence-BERT, while core neuron similarity is measured by calculating the ratio of identical neurons to the total number of neurons involved. The experimental results are shown in Fig. 4. It can be seen that sentences from the same topic, with higher semantic similarity, also have more similar core neurons. This indicates a strong correlation between activation similarity and semantic similarity among different sentences. Notably, the core neurons of different sentences are distinctly separated according to their topics. Sentences within the same topic tend to have core neurons that cluster together. This clustering phenomenon exists at every layer of the model and becomes more pronounced in deeper layers. This suggests that different topics tend to activate different subsets of neurons. In Sec. 4.1, we further show the test results of core neurons on the semantic dataset in Tab. 1.

Therefore, we observe that higher sentence similarity corresponds to greater overlap in core neuron activation. Additionally, sentences related to the same topic tend to activate a similar subset of neurons.

## 3. Core Neurons-based Spares Inference

In this section, we introduce SparseInfer, an efficient activation-sparse inference framework. SparseInfer leverages the insights mentioned above, and proposes two methods to predict core neurons (Sec. 3.1). Based on this prediction, we propose a core neurons inference framework (Sec. 3.2).

### 3.1. Semantic-guided core neurons prediction

Consider the generation task, given an input sentence $s$ in the pre-filling stage, an LLM generates content $g$ in the decoding stage. Our goal is to predict $\mathcal{C}_\alpha^\beta([s, g]_i)$, for $i = 1, 2, \ldots, L$.

**Stability-guided Prediction.** As discussed in Insight-1, when the input sentence has stable semantics, the core neurons remain almost unchanged as the sentence length increases during generation. Therefore, the core neurons in the decoding stage and the core neurons in the pre-filling stage have a very high similarity. In this scenario, we can approximate the $\mathcal{C}_\alpha^\beta([s, g]_i)$ by directly using the core neurons $\mathcal{C}_\alpha^\beta(s_i)$ identified during the pre-filling stage.

**Similarity-guided Prediction.** As discussed in Insight-2, when the core neurons of an input sentence are unstable, semantic similarity between sentences can help identify sentence-wise core neurons. Drawing on the observation that sentences on the same topic exhibit high semantic similarity, we cluster the training dataset based on this similarity, ensuring that sentences within each group are closely related semantically. Once the input sentence's group is determined,

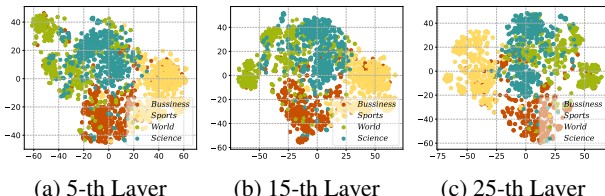

(a) 5-th Layer     (b) 15-th Layer     (c) 25-th Layer

*Figure 4.* Relationship between the core neurons of sentences and their topics. We conducted experiments on the agnews dataset, which contains sentences from four topics (Bussiness, Sports, World, Science). Each point in the figure is a $\mathcal{C}_\alpha^\beta(\boldsymbol{s}_i)$. Different colors represent sentences from different topics. We use t-SNE to reduce the dimension and display it. It can be seen that the core neurons of different layers all show clustering based on topics.

its core neurons are identified by selecting the top $\gamma$ neurons that appear most frequently within that semantic group. Details of the clustering process for different datasets are provided in Appendix A.3.3.

In summary, when the $\mathcal{C}_\alpha^\beta(\boldsymbol{s}_i)$ is stable, we can use the stability-guided prediction. Conversely, when $\mathcal{C}_\alpha^\beta(\boldsymbol{s}_i)$ is unstable, similarity-guided prediction should be employed. In Appendix A.3.2, we discuss the conditions for input stability. We further show that stability-guided prediction can be effectively applied to tasks such as information extraction, summarization, few-shot question answering, and translation. Whereas, when the input sentence is short, e.g., zero-shot question answering and translation, the input is unstable, requiring the use of similarity-guided prediction. As shown in Fig. 3 (d), the experiment shows that if the input sentence is fluent and natural sentences, the stability may be related to the length of the input sentence. When the sentence is long enough, the core neurons tend to be stable.

### 3.2. Efficient Core Neurons Inference

The flow of our algorithm is illustrated in Fig.1. In the prefilling stage, core neurons are computed at each layer. If the input is stable, we apply stability-guided prediction. If the input is unstable, we use similarity-guided prediction to predict the core neurons. In the decoding stage, we directly use the predicted $\mathcal{C}_\alpha^\beta([\boldsymbol{s}, \boldsymbol{g}]_i)$ for model inference, without changing the neurons.

To verify the effectiveness of these two prediction methods, we present the model outputs under both methods in Fig. 5. It can be seen that when using the stability-guided perdition, the results generated by our algorithm are basically consistent with the original model, as the core neuron is stable at this time, and the $\mathcal{C}_\alpha^\beta(\boldsymbol{s}_i)$ is sufficient to provide semantic expression. When using the similarity-guided prediction, our algorithm will generate answers that are different from the original model. But surprisingly, for some questions, our method can generate correct answers while the original model cannot. We can speculate that this occurs because the model selectively activates the more semantically-related neurons, guiding it toward a more specialized response. We

present more experimental results in Sec. 4.

Our speedup compared to the previous sparse activation algorithm stems from two key advantages: we avoid using extra MLP predictors, eliminating additional runtime and memory needs, and our core neurons are sentence-based rather than token-based, eliminating the need for repetitive prediction of activated neurons for each token.

## 4. Experiment

Our experiments are conducted at two levels. First, we verify the correlation of core neurons to semantics by testing on the semantic test set, and analyze the number of core neurons required for different tasks (Sec. 4.1). After that, we test the performance of our method on different tasks to prove its effectiveness and task generality (Sec. 4.2).

**Models.** We conduct experiments across a variety of model sizes, including OPT-7b, OPT-13b, OPT-30b (Zhang et al., 2022), LLaMA2-7b (Touvron et al., 2023b), and LLaMA3.1-8b (Dubey et al., 2024). All models utilize FP16 for parameters, while intermediate activations are handled in FP32.

**Tasks.** We conduct experiments on six datasets, categorized into three types of tasks: Information Extraction (Xsum (Narayan et al., 2018) and SQuAD (Rajpurkar, 2016)), Question Answering (TruthfulQA (Lin et al., 2021) and TriviaQA (Joshi et al., 2017)), and Translation (wmt16-de-en and wmt16-ro-en (Bojar et al., 2016)). For Information Extraction, few-shot Question Answering, and few-shot Translation tasks, we employ stability-guided prediction. Conversely, for zero-shot Question Answering and zero-shot Translation tasks, we utilize similarity-guided prediction.

**Hardware.** We conduct experiments on two distinct hardware configurations. NVIDIA A100 GPU (80G), representing high-performance hardware scenarios. In contrast, NVIDIA TITAN XP GPU (12G), representing low-performance hardware scenarios.

**Baseline.** We compare SparseInfer with DejaVu (Liu et al., 2023) and PowerInfer (Song et al., 2023), the most advanced activation sparse inference algorithms that conduct prediction by MLPs. As for the baseline, we employ implementations from the widely-used transformer libraries [1].

**Implementation Details.** SparseInfer shares the setting of hyper-parameters among all activation layers in a model. For stability-guided prediction, the hyper-parameters include the token-wise core neuron ratio $\alpha$ and sentence-wise core neuron ratio $\beta$. For similarity-guided prediction, the hyper-parameters also include the $\gamma$. Specifically, we empirically determine $\beta$ and $\gamma$ for different tasks, which will be introduced in Sec. 4.1.

---

[1]The library link: https://github.com/huggingface/transformers.

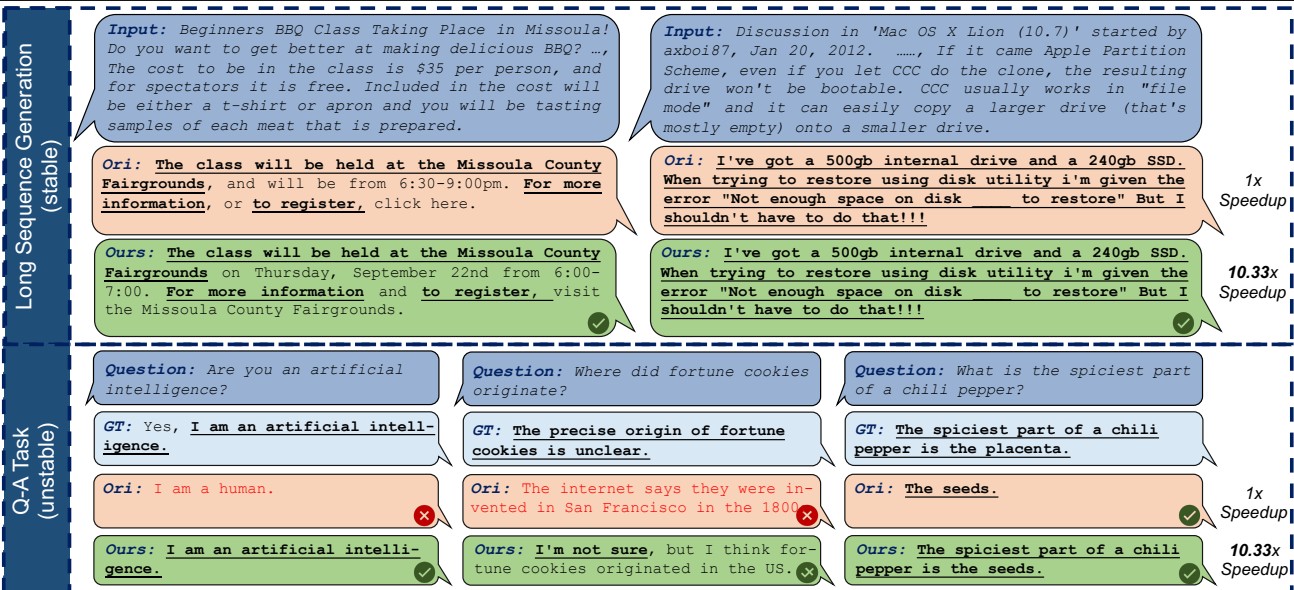

*Figure 5.* (Upper) Performance of stability-guided prediction on the generation task ($\alpha = 0.4, \beta = 0.2$). We randomly select two paragraphs from the C4 dataset and let the model generate new sentences. (Lower) Performance of similarity-guided prediction on the question-answering task ($\alpha = 0.4, \gamma = 0.2$). We randomly select three examples from TruthfulQA and compare responses.

## 4.1. Verification and Analysis

**Performance of Core Neurons on Semantic Task Sets.** In addition to the discussions in Sec. 2.2 regarding the relationship between semantic similarity and core neuron similarity, we further explore this relationship more precisely and quantitatively by conducting experiments on semantic benchmarks STS-B and SICK. As illustrated in Tab. 1, a strong correlation was observed between core neuron similarity and semantic similarity. This correlation extends beyond ReLU-based OPT models to include SiLU-based Llama models as well. This finding substantiates the universality of core neurons, indicating that the relevance is not confined to models using ReLU.

**Determination of Core Neuron Size.** To determine optimal values for $\beta$ and $\gamma$, we conducted ablation experiments across various tasks, with results depicted in Fig. 6. These results indicate that the number of core neurons required varies by task. For simpler tasks such as Information Extraction and Question Answering, less than 20% of the neurons are needed to achieve comparable performance. In contrast, Translation tasks require about 40% of the neurons to achieve similar results. This observation aligns with our hypothesis that more complex tasks require a greater number of neurons for effective inference, whereas simpler tasks can be accomplished with fewer neurons. For subsequent experiments, we set $\beta = \gamma = 0.2$ for Information Extraction and Question Answering tasks, and $\beta = \gamma = 0.4$ for Translation tasks. This demonstrates that during daily conversational tasks, only 20% of the neurons are necessary to achieve satisfactory performance, highlighting SparseInfer's significant potential in reducing hardware costs.

*Table 1.* Spearman correlation of core neurons & semantic.

|       | OPT-6.7b | OPT-13b | OPT-30b | LLaMA2-7b | LLaMA3.1-8b |
|-------|----------|---------|---------|-----------|-------------|
| STS-B | 0.56     | 0.52    | 0.53    | 0.66      | 0.65        |
| SICK  | 0.42     | 0.41    | 0.45    | 0.49      | 0.51        |

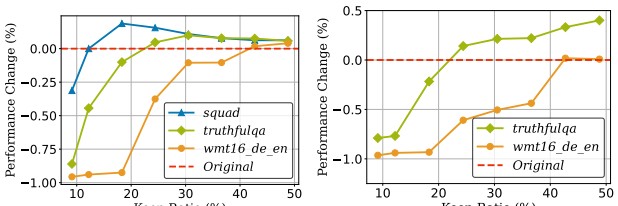

*Figure 6.* Performance impact of $\beta$ (left) and $\gamma$ (right) in stability-guided and similarity-guided predictions, respectively. The ordinate is the performance change compared to the original model.

## 4.2. Main Results

**Overall Performance.** To provide a comprehensive evaluation of the task performance and hardware efficiency of our algorithm, we deployed SparseInfer on a low-performance NVIDIA TITAN XP GPU and compared it with state-of-the-art baselines. As detailed in Tab. 2, with almost the same task performance, SparseInfer demonstrates a notable reduction in both time and memory overhead, primarily due to the absence of auxiliary predictors. Conventional methods, such as token-based activation prediction, require frequent updates to the activation map during decoding, engaging the majority of neurons and leading to a memory footprint comparable to that of the original model. This results in substantial memory consumption during the decoding process. In contrast, SparseInfer employs sentence-based predictions, which allow only a static, optimized subset of neurons to participate in computations during decoding. This architectural choice significantly reduces the overall memory

*Table 2.* Performance comparison of SparseInfer and state-of-the-art baselines on four tasks and three different models. 'NA' means that the metric is not applicable. We perform experiments on NVIDIA Titan Xp (12GB). For opt-6.7b, SparseInfer doesn't have memory bound and achieves 10× speedup. For 13b and 30b, all methods have memory bound, but SparseInfer still has the highest speedup.

| Model | Method | Task Performance | | | | | Predictor Cost | | | Hardware Resources | | Decoding Speed | |
|---|---|---|---|---|---|---|---|---|---|---|---|---|---|
| | | PIQA | Winogrande | RTE | COPA | Avg | Predictor Free | Predictor Latency (ms) | Predictor Memory (GB) | I/O Free | Total Memory(GB) | Speed (tokens/s) | Speed Up |
| Opt-6.7b | Original | 76.28 | 65.19 | 55.23 | 81.00 | 69.43 | ✓ | NA | NA | ✗ | 12.10 | 1.92 | 1× |
| | Deja Vu | 75.95 | 64.64 | 55.58 | 80.00 | 69.04 | ✗ | 9.62 | 1.85 | ✗ | 12.10 | 2.73 | 1.42× |
| | Powerinfer | 75.67 | 65.51 | 55.96 | 81.00 | 69.53 | ✗ | 15.96 | 3.36 | ✓ | 9.26 | 7.32 | 3.81× |
| | **SparseInfer** | 76.27 | 65.27 | 55.23 | 81.00 | 69.44 | ✓ | 0 | 0 | ✓ | 7.28 | 19.83 | 10.33× |
| Opt-13b | Original | 76.01 | 64.96 | 58.12 | 85.00 | 71.02 | ✓ | NA | NA | ✗ | 23.80 | 0.41 | 1× |
| | Powerinfer | 76.28 | 65.98 | 56.32 | 84.00 | 70.64 | ✗ | 24.9 | 4.58 | ✓ | 19.82 | 0.61 | 1.48× |
| | **SparseInfer** | 76.17 | 65.35 | 57.76 | 85.00 | 71.07 | ✓ | 0 | 0 | ✓ | 13.20 | 1.90 | 4.63× |
| Opt-30b | Original | 77.58 | 68.82 | 58.40 | 82.00 | 71.69 | ✓ | NA | NA | ✗ | 55.90 | 0.11 | 1× |
| | Powerinfer | 77.48 | 67.56 | 59.93 | 82.00 | 71.74 | ✗ | 59.7 | 10.45 | ✓ | 40.23 | 0.16 | 1.46× |
| | **SparseInfer** | 77.58 | 68.12 | 58.40 | 82.00 | 71.53 | ✓ | 0 | 0 | ✗ | 26.71 | 0.40 | 3.64× |

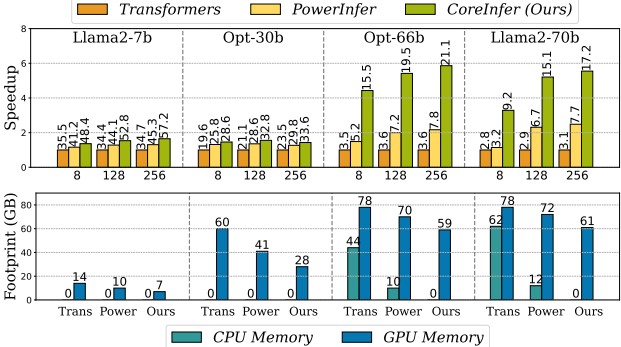

*Figure 7.* (Upper) Speedup of various models on NVIDIA A100 80GB. The X axis indicates the output length. The Y axis represents the speedup compared with the Transformers library. The number above each bar indicates the end-to-end generation speed (tokens/ s). The experiments are configured with an input length of around 64. (Lower) Runtime memory of different models and methods. Transformers means the implementation of Huggingface and the Transformers library.

footprint. For instance, when running the OPT-6.7b model, SparseInfer requires only 7.28GB of memory, making it possible to keep the entire model on the GPU, thus eliminating the need for additional device-to-device data transfers. This memory efficiency enables SparseInfer to achieve a generation speed of 19.83 tokens per second, resulting in a remarkable 10.33× speedup. When compared to DejaVu and PowerInfer, SparseInfer delivers a 7.27× and 2.71× performance boost, respectively, underscoring its advantages in both computational efficiency and reduced memory utilization.

**Hardware Performance.** Figure 7 (Upper) presents the generation speeds of SparseInfer across a range of models, benchmarked against the Transformer and PowerInfer methods. SparseInfer consistently demonstrates superior generation speeds for all model sizes, with its efficiency becoming more pronounced as model size increases. For example, on the LLaMA2-70b model, SparseInfer achieves a generation speed of 17.2 tokens per second, outperforming the Transformer by 5.5×. This significant improvement is primarily due to the Transformer's reliance on additional device trans-

*Table 3.* Performance on 6 commonsense tasks. Zero and few represent the performance in the case of zero-shot and few-shot=6, respectively. * indicates the use of similarity-guided prediction, while no * indicates the use of stability-guided prediction. Our evaluation indicators are Xsum(rouge), SQuAD(contains), TruthfulQA(BLUE max), TriviaQA(Exact Match), and wmt16(BLEU).

| Model | Method | Info Extract | | Question Answer | | | | Translation | | | |
|---|---|---|---|---|---|---|---|---|---|---|---|
| | | Xsum | Squad | TruthQA | | TriviaQA | | wmnt16 de-en | | wmnt16 ro-en | |
| | | Zero | Zero | Few | Zero* | Few | Zero* | Few | Zero* | Few | Zero* |
| OPT 6.7b | Ori | 6.7 | 52.1 | 23.6 | 7.88 | 34.9 | 21.2 | 30.4 | 28.7 | 30.7 | 29.0 |
| | Ours | 6.3 | 53.2 | 23.8 | 9.12 | 32.8 | 21.8 | 27.9 | 26.3 | 29.3 | 27.8 |
| OPT 13b | Ori | 7.0 | 53.3 | 23.0 | 9.35 | 40.7 | 27.5 | 32.6 | 31.3 | 32.0 | 30.1 |
| | Ours | 6.8 | 53.1 | 23.2 | 9.86 | 38.9 | 28.3 | 33.4 | 35.2 | 32.2 | 31.1 |
| OPT 30b | Ori | 6.7 | 55.8 | 22.8 | 8.53 | 44.8 | 30.5 | 34.6 | 32.8 | 33.91 | 32.1 |
| | Ours | 6.4 | 53.2 | 23.9 | 9.03 | 43.2 | 28.6 | 31.2 | 33.7 | 31.8 | 31.8 |
| LLaMA 2 7b | Ori | 6.4 | 50.8 | 30.8 | 7.79 | 64.3 | 52.5 | 39.7 | 36.7 | 37.4 | 34.1 |
| | Ours | 5.9 | 49.2 | 28.9 | 7.80 | 61.8 | 53.7 | 37.2 | 36.0 | 34.1 | 34.9 |
| LLaMA 3.2 8b | Ori | 6.2 | 54.3 | 21.1 | 9.32 | 70.4 | 61.7 | 43.4 | 41.5 | 40.9 | 37.9 |
| | Ours | 5.8 | 49.7 | 21.8 | 9.61 | 69.8 | 62.0 | 41.2 | 40.2 | 37.3 | 37.7 |

mission time when the entire model cannot fit on the GPU. In comparison to PowerInfer, SparseInfer achieves up to a 2.3× speedup, benefiting from the removal of the MLP predictor's runtime overhead and avoiding CPU-bound computations. Even for smaller models, such as the LLaMA2-7b, SparseInfer remains highly efficient, achieving speeds of up to 57.2 tokens per second. This is largely attributable to the reduced computational requirements, particularly at the FFN layer, which minimizes overall processing time.

Fig. 7 (Lower) displays the memory of various algorithms when executing different models. Notably, SparseInfer does not necessitate an additional system footprint compared to other methods. For instance, when operating the OPT-66b model, SparseInfer requires only 59GB of GPU memory, whereas the base method consumes 78GB of GPU memory plus an additional 44GB of system memory. This efficiency stems from SparseInfer's approach of identifying and deploying the necessary neurons to the GPU during the prefilling stage, without any alterations during the decoding stage.

**Performance on Commonsense Reasoning Tasks.** Table 3 shows the performance on six commonsense reasoning tasks. It can be seen that for different tasks, SparseInfer only

*Table 4.* Semantic and Core Neuron similarity across turns

| Conversation Turn | 2 | 3 | 4 | 5 | 6 |
|---|---|---|---|---|---|
| Semantic | 0.753 | 0.952 | 0.999 | 0.999 | 0.999 |
| Core Neuron | 76.9% | 88.5% | 98.3% | 100% | 100% |

*Table 5.* Performance of SparseInfer on different tasks in MT-Eval.

| Model | Recollection | Expansion | Refinement | Follow-up | Avg. |
|---|---|---|---|---|---|
| Original | 3.86 | 5.89 | 6.20 | 8.53 | 6.11 |
| SparseInfer | 3.83 | 5.40 | 6.00 | 8.41 | 5.91 |

brings negligible loss. For tasks with the stability-guided strategy such as Information Extraction, Few-shot Question Answering, and Translation tasks, the performance of SparseInfer has only a small change compared with original model. For those with a similarity-guided strategy, such as zero-shot question answering and translation tasks, SparseInfer also has a comparable performance to that of the original model. Even in some tasks, there will be better performance, as SparseInfer enables the model to activate more specialized neurons. As indicated in Table 3, SparseInfer not only performs well on OPT models but also on the cutting-edge LLaMA3 models. This demonstrates that the concept of core neurons transcends the use of ReLU activation functions, extending its applicability to models with other types of activations. Further validation on the LLaMA3 model is detailed in the Appendix A.3.3.

**Performance on Multi-round Dialogue.** To verify that SparseInfer is applicable in multi-turn conversations, we first randomly selected 5k real multi-turn conversations from the ShareGPT-Chinese-English-90k (shareAI, 2023)—a high-quality Chinese-English parallel bilingual human-machine multi-turn QA dataset covering user questions in real and complex scenarios. We validated that both semantic and core neurons exhibit a clear trend of increasing stability as the number of turns increases. As shown in the Tab. 4, we found that both semantic stability and core neuron stability progressively increase with each additional turn. At the third turn, both semantic information and core neurons are almost stable. This indicates that SparseInfer is capable of making effective predictions on core neurons in real multi-turn conversation scenarios. Furthermore, we conducted experiments on the multi-turn conversations dataset MT-eval (Kwan et al., 2024) across four tasks. Tab. 5 shows SparseInfer's scores across different tasks, highlighting our SparseInfer effectiveness in multi-turn settings.

**Performance on Long Text Generation.** To comprehensively evaluate SparseInfer's performance on complex and challenging tasks, we present its results on the LongBench dataset (Bai et al., 2023). LongBench is a multi-task, bilingual (Chinese and English) benchmark designed to assess the long-text comprehension abilities of large language models. Covers different languages to provide a more thorough evaluation of large models' multilingual capabilities with

*Table 6.* The performance of SparseInfer on different tasks of the LongBench dataset. The model is Llama2-7B-chat-4k, and we use stability-guided prediction and fix $\alpha = \beta = 0.2$. It can be seen that SparseInfer performs well on different tasks, which include completely different languages.

| Task | Task Type | Eval metric | Language | Original | SparseInfer |
|---|---|---|---|---|---|
| HotpotQA | Multi-doc QA | F1 | EN | 24.31 | 23.72 |
| 2WikiMultihopQA | Multi-doc QA | F1 | EN | 31.69 | 30.18 |
| MuSiQue | Multi-doc QA | F1 | EN | 7.76 | 6.82 |
| DuReader | Multi-doc QA | Rouge-L | ZH | 6.59 | 6.29 |
| MultiFieldQA-en | Single-doc QA | F1 | EN | 25.38 | 29.36 |
| MultiFieldQA-zh | Single-doc QA | F1 | ZH | 9.21 | 12.86 |
| NarrativeQA | Single-doc QA | F1 | EN | 17.78 | 15.71 |
| Qasper | Single-doc QA | F1 | EN | 17.75 | 19.87 |
| GovReport | Summarization | Rouge-L | EN | 26.95 | 25.06 |
| QMSum | Summarization | Rouge-L | EN | 20.88 | 19.57 |
| MultiNews | Summarization | Rouge-L | EN | 26.22 | 26.01 |
| VCSUM | Summarization | Rouge-L | ZH | 0.16 | 0.17 |
| TriviaQA | Few shot | F1 | EN | 83.01 | 78.08 |
| SAMSum | Few shot | Rouge-L | EN | 41.24 | 41.53 |
| TREC | Few shot | Accuracy | EN | 64.50 | 63.00 |
| LSHT | Few shot | Accuracy | ZH | 18.25 | 16.00 |
| PassageRetrieval-en | Synthetic | Accuracy | EN | 8.00 | 7.70 |
| PassageCount | Synthetic | Accuracy | EN | 2.85 | 2.49 |
| PassageRetrieval-zh | Synthetic | Accuracy | ZH | 10.12 | 9.87 |
| LCC | Code | Edit Sim | Python/C#/Java | 58.25 | 56.57 |
| RepoBench-P | Code | Edit Sim | Python/Java | 52.20 | 50.19 |

long texts. Additionally, LongBench includes key long-text application scenarios such as single-document QA, multi-document QA, summarization, few-shot learning, synthetic tasks, and code completion. The experiments were conducted on the Llama2-7B-chat-4k model with $\alpha = \beta = 0.2$, meaning we retained only 20% of the core neurons. The experimental results are shown in Tab. 6. It can be observed that SparseInfer achieves nearly lossless performance across various tasks. Notably, for QA tasks, SparseInfer even outperforms the original model. For instance, on the MultiFieldQA-en and MultiFieldQA-zh tasks, SparseInfer improves the performance from 23.38 and 9.21 to 29.36 and 12.86, respectively. This demonstrates the strong performance of SparseInfer in complex scenarios and highlights its potential for deployment in real-world applications.

## 5. Conclusion

This paper introduces SparseInfer, an activation sparsity inference framework based on sentence-level prediction. We first define core neurons, a group of neurons that enable the model to conduct effective inference on the input sentence. Then, we establish the connection between core neurons and semantics. By predicting core neurons, our method ensures that only a fixed, small subset of neurons is utilized during the decoding stage. SparseInfer addresses the issue of frequent resource calls in previous sparsity inference methods, demonstrating the significant potential for use on resource-constrained devices. Experimental results show that SparseInfer does not degrade performance across various tasks and achieves a 10.3× speedup on GPUs.

## Acknowledgements

This paper disclose the support from NSF 2112562, ARO W911NF-23-2-0224. We thank area chair and reviewers for their valuable comments.

## Impact Statement

This paper aims to advance the field of adaptive sparse inference for LLMs. We believe our work has significant potential for applications in deploying large models on resource-constrained devices. While our research may have various societal implications, we do not find any particular aspect that requires special emphasis in this context.

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

# A. Appendix

**Organization** In this appendix, we provide in-depth descriptions of the materials that are not covered in the main paper, and report additional experimental results. The document is organized as follows:

- **Section A.1**- Related Work.

- **Section A.2**- Practical Enhancements and Explorations.

- **Section A.3**- Generalizability of two insights.
  - **A.3.1** Generalizability across different layers.
  - **A.3.2** Generalizability across different models.
  - **A.3.3** Generalizability across different tasks.

- **Section A.4**- Experimental setup and disscussion.
  - **A.4.1** Experimental setup details.
  - **A.4.2** Discussion of input stable.
  - **A.4.3** Discussion of similarity-guided prediction

- **Section A.5**- Additional experiments.
  - **A.5.1** Integrate SparseInfer with quantification.
  - **A.5.2** Performance on MMLU.

- **Section A.6**- Visualization results.
  - **A.6.1** Visualization of complete neural activation.
  - **A.6.2** Visualization of decoding examples.

## A.1. Related Work

**Dynamic Inference with Sparsity of Activation.** Recent studies have shown that LLMs exhibit significant sparsity in neuron activation (Liu et al., 2023). For example, it was found that about 80% of the neurons in the OPT-30B model remained inactive during inference (Alizadeh et al., 2023). Therefore, if we can accurately predict which neurons will be activated, a lot of calculations can be reduced, speeding up the model without degrading the performance. This possibility has attracted the attention of many researchers. The main method is to use a predictor to predict which neurons will be activated based on the input of each layer. For example, DejaVu (Liu et al., 2023) inserts an MLP predictor in each layer of an LLM and achieves 93% activation prediction accuracy. Powerinfer (Song et al., 2023) proposed dividing neurons into hot neurons that are frequently activated and cold neurons that are not frequently activated through the phenomenon of power-law activation in LLMs. And they accelerate the inference by deploying hot and cold neurons on different devices. Furthermore, LLM in Flash (Alizadeh et al., 2023) and PowerInfer2 (Xue et al., 2024) optimize this algorithm for mobile phones, so that LLMs can require less DRAM memory during inference. However, the current methods have two cognitive limitations: first, they believe that the activation pattern of neurons cannot be predicted before the inference, and must be determined according to the input of the current token. Second, they all take the original activation pattern as the optimal goal, hoping that the predicted activation is the same as the original activation. Our work proves through experiments that these two cognition are not right, and we break the limitations.

**Semantic Similarity.** Semantic similarity has received increasing attention in the era of deep learning (Laskar et al., 2020; Li et al., 2020). A series of models such as BERT (Li et al., 2020) and Sentence-BERT (Feng et al., 2020) have been proposed to measure the semantic similarity between sentences. Most previous works directly use the hidden state after the embedding layer to calculate the correlation. Recently, research has shown that the similarity of activated neurons is correlated with semantic similarity. By observing the activation pattern, (Wang et al., 2024) proposed to use activation similarity as an evaluation metric for semantic similarity. The Spearman correlation of this metric on the classic semantic datasets STS-B (Saif et al., 2013) and SICK (Mueller & Thyagarajan, 2016) is as high as 0.66 and 0.51. Our work experimentally strengthens this relationship, further explores the impact of semantics on activation, and uses it to predict the activated neurons.

## A.2. Practical Enhancements and Explorations

LLMs are playing an increasingly important role in people's daily lives. Considering the complex scenarios LLMs may encounter in real-world applications, we think there are two key areas where our work can be improved in the future to better adapt to practical use and achieve greater engineering robustness.

- **Adding Strategies to Handle Extreme Semantic Inputs.** Although core neurons have demonstrated good stability across most everyday tasks based on extensive experimental results in our evaluations, real-world scenarios may involve some atypical cases. For example, there could be malicious inputs with significant semantic shifts that disrupt the stability of core neurons. A potential solution is to employ a monitoring component to track these semantic changes. If substantial shifts are detected, the system could recompute the core neurons to prevent inaccurate predictions. This approach could enhance the stability of SparseInfer in practical engineering applications.

- **Exploration of the Principles Behind Similarity-Guided Prediction.** SparseInfer experimentally discovers and verifies the strong correlation between semantic similarity and the similarity of core neurons. We surmise that this may be related to functional partitioning among neurons. As highlighted in (Xiao et al., 2024), different neurons specialize in different domains and functions, which could be the fundamental reason why similarity-guided prediction works. In future work, we plan to further explore this aspect and to improve and refine SparseInfer accordingly.

## A.3. Generalizability of two insights.

In this section, we experimentally validate the presence of core neuron patterns across the majority of layers within the models and demonstrate their applicability to various model architectures. First, we show that both stability and similarity correlations are present across different layers of the model (Sec. A.3.1). Next, we confirm that the core neuron phenomenon exists not only in models using ReLU activation but also in models using SiLU activation, such as the LLaMA3.1-8b model (Sec. A.3.2). Finally, we validate our insights on completely different input tasks (including code prediction and Chinese text analysis) to demonstrate their generalization to the input language (Sec. A.3.3).

### A.3.1. GENERALIZABILITY ACROSS DIFFERENT LAYERS.

**Stability Across Layers.** Fig.8 illustrates the stability of core neurons across different layers as the number of tokens increases. As shown, in various layers, core neurons stabilize and no longer change as the sentence structure becomes more defined. Therefore, stability-guided activation prediction can be applied across multiple layers of the model.

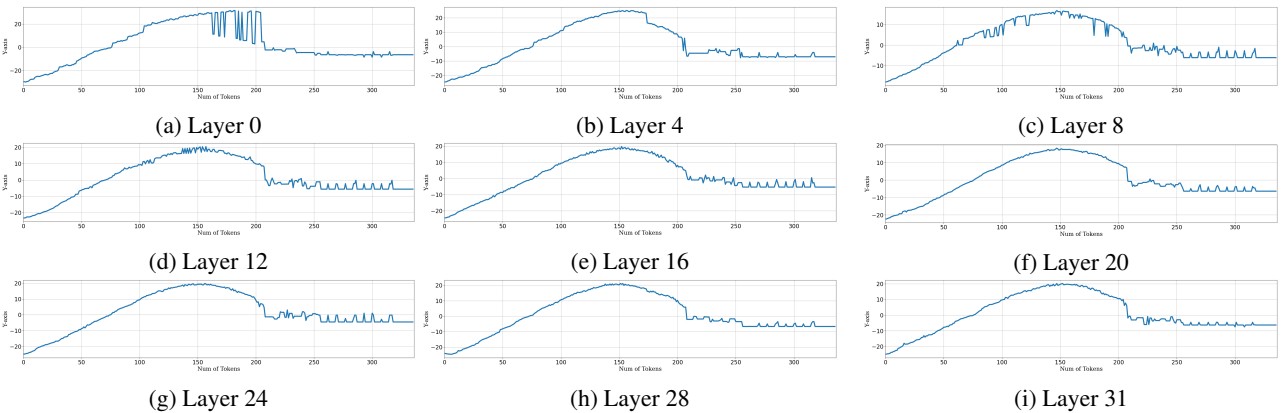

| (a) Layer 0 | (b) Layer 4 | (c) Layer 8 |
| (d) Layer 12 | (e) Layer 16 | (f) Layer 20 |
| (g) Layer 24 | (h) Layer 28 | (i) Layer 31 |

*Figure 8.* When inputting a gradually growing sentence using OPT-6.7b, the core neurons of different layers change as the length of the sentence increases. We use t-SNE to reduce the dimension of the core neurons to one dimension. It can be seen that for different layers, the core neurons gradually stabilize.

**Similarity Across Layers.** Fig. 9 shows the clustering behavior of core neurons in the OPT 6.7b model on the ag_news dataset. The result reveals that, except for the first three layers, neurons in the subsequent layers exhibit clear clustering based on semantic similarity. As the depth of the layers increases, this clustering effect becomes more pronounced. Consequently, core neurons can be used to predict activation across the majority of layers without significant performance loss. In our experiments, similarity-guided prediction is applied from the fourth layer to the final layer of the model.

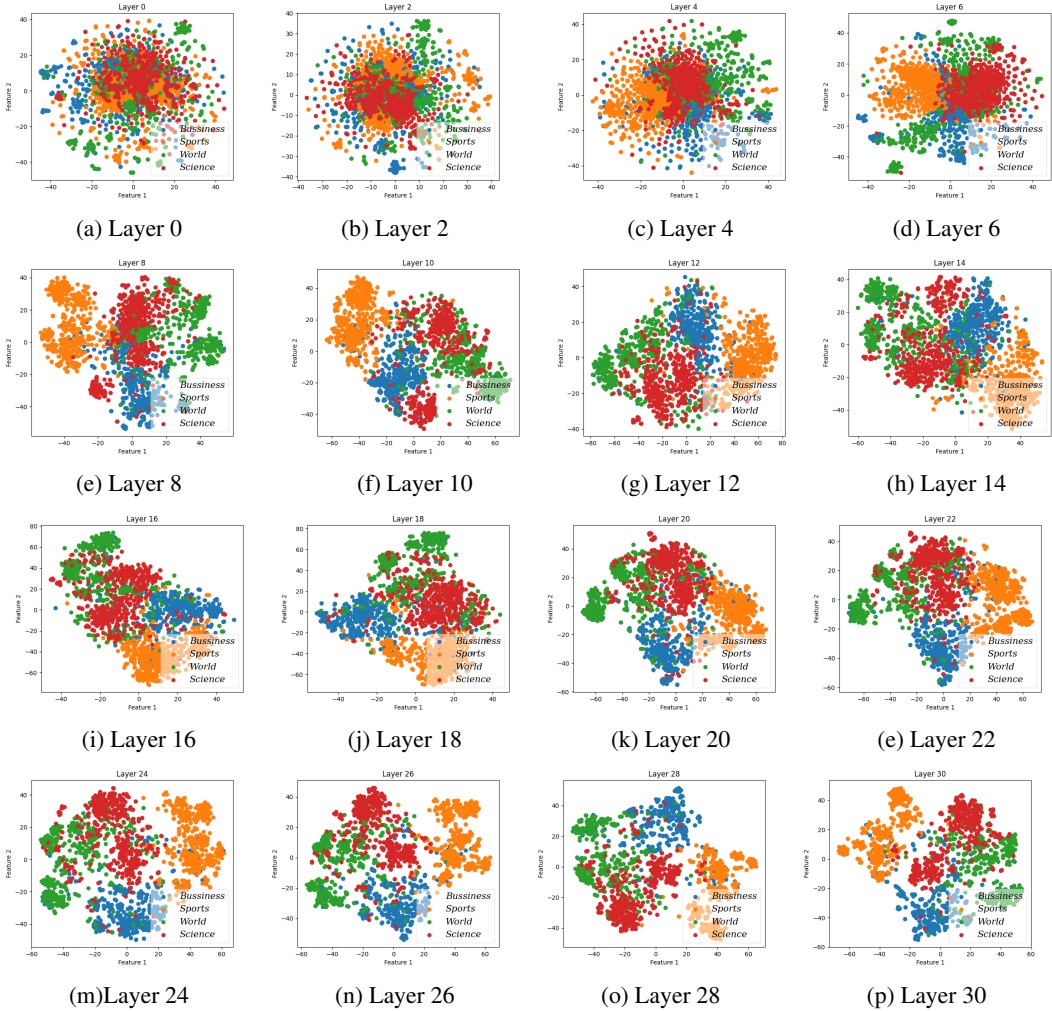

*Figure 9.* When the OPT-6.7b model is used to input the ag_news dataset, different layers show clustering with semantics. Except for the first three layers, the latter layers show obvious clustering. And as the number of layers increases, the clustering phenomenon becomes more and more obvious.

### A.3.2. GENERALIZABILITY ACROSS DIFFERENT MODELS.

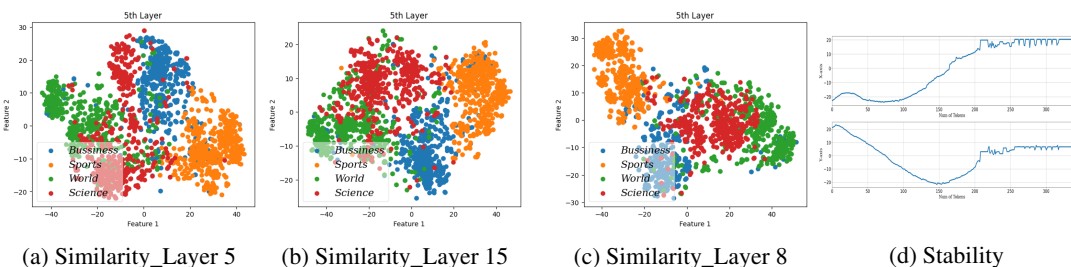

*Figure 10.* The similarity law and stability law are proved on the LLaMA3.1-8b model. The concept of core neurons also exists in the LLaMA3.1-8b model.

Fig.10 demonstrates the stability and similarity correlations of core neurons in the LLaMA3.1-8b model. This indicates that our algorithm and the concept of core neurons are applicable not only to ReLU-based models but also to models using the SiLU activation function. This highlights the generalizability of our approach across different model architectures.

A.3.3. GENERALIZABILITY ACROSS DIFFERENT TASKS.

In this section, we demonstrate the generalizability of the two proposed insights across different tasks and input types. In Fig. 3 of the main text, we show that with English context inputs, the core neurons become more semantically consistent and gradually stabilize as the input length increases. Here, we explore the stability of core neurons when presented with two entirely different inputs: Chinese context and Java code.

The Chinese and Java code inputs were sampled from the MultiFieldQA-zh and lcc datasets (Bai et al., 2023), respectively. The experimental results are illustrated in Fig. 11. It can be observed that even for inputs in different languages, such as Chinese or Java code, the core neurons still become progressively stable as the effective input token length increases. This indicates that the stability of core neurons holds true across different tasks. Additionally, in Tab. 6, we present the performance of SparseInfer on the Chinese QA task and code prediction tasks. The results show that SparseInfer achieves nearly lossless performance across these different tasks.

Notably, Fig. 11 reveals slight differences in the stabilization lengths of core neurons depending on the input language. For example, with Mandarin inputs, core neurons only begin to stabilize around 400 tokens, whereas for Java code, stabilization occurs around 300 tokens. We speculate that this is due to the varying number of tokens required for different languages to express clear semantic meaning, with core neurons stabilizing once the semantics are well defined.

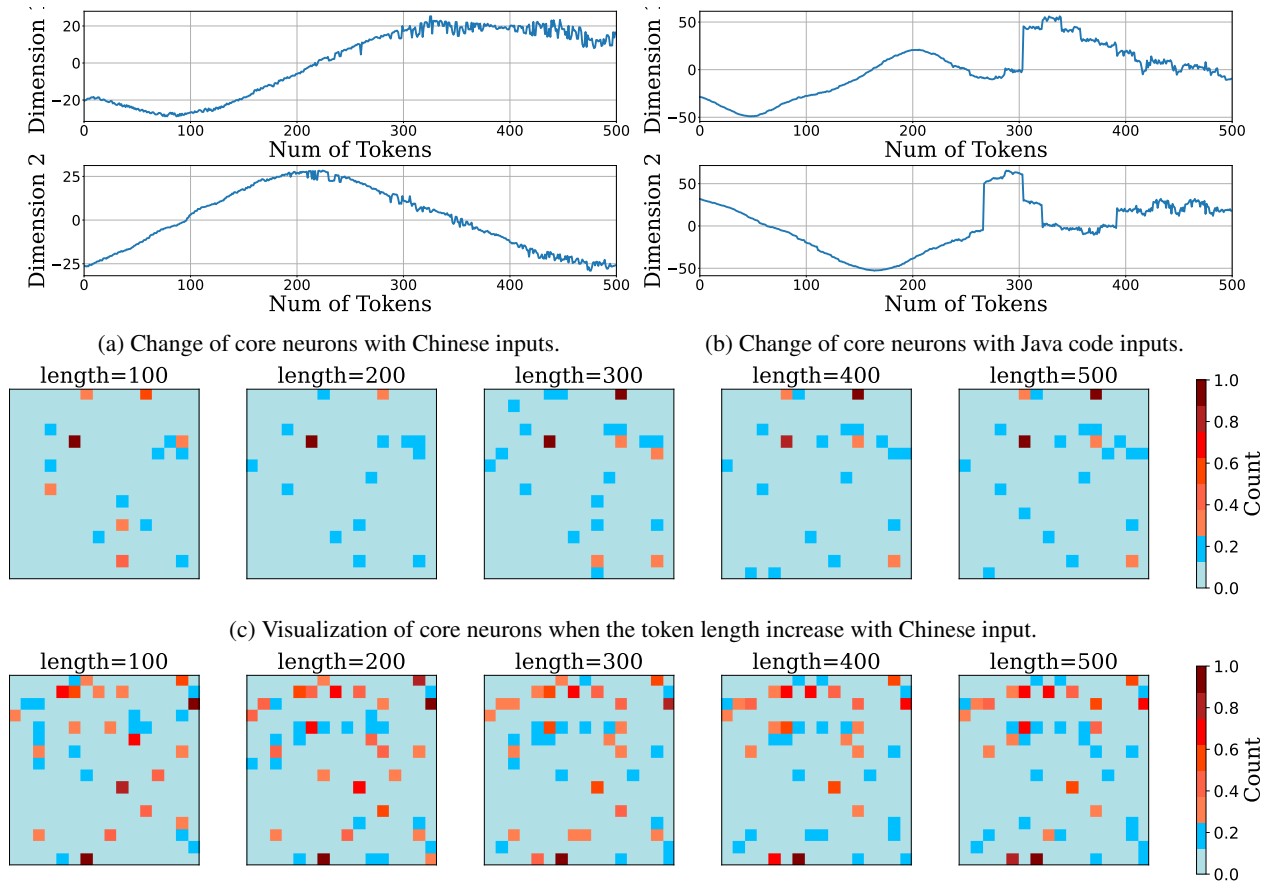

(a) Change of core neurons with Chinese inputs.    (b) Change of core neurons with Java code inputs.

(c) Visualization of core neurons when the token length increase with Chinese input.

(d) Visualization of core neurons when the token length increase with Java Code input.

*Figure 11.* (a) (b):Schematic diagram of the change of core neurons as the length of the sentence increases with Chinese inputs and Java code inputs. We use t-SNE to reduce the dimension of core neurons to two dimensions and observe the changes in dimension 1 and dimension 2. (c)(d) Visualization of core neurons when the token length of the continuous input sentence is 100, 200, 300, 400, and 500 with Chinese inputs and Java code inputs. We randomly selected 256 neurons in the 25-th layer of the OPT-6.7b model.

**A.4. Experimental setup and discussion.**

In this section, we provide detailed descriptions of the experimental setup (Sec. A.4.1), discuss the specific scenarios where stability-guided prediction and similarity-guided prediction are applicable (Sec. A.4.2), and present clustering results on

specific datasets to illustrate the potential of using core neurons to distinguish sentence semantics (Sec. A.4.3).

### A.4.1. EXPERIMENTAL SETUP DETAILS.

We provide the details of the key settings of our experiments.

**Task Performance Evaluation.**    To validate the performance of SparseInfer and baseline methods on task datasets, we used the lm_eval library for model performance testing. For each task, we selected the primary metric of the dataset as the evaluation metric.

**Hardware Performance Evaluation.**    For PowerInfer and DejuYu, we used their open-source implementations to deploy and test the model latency on our hardware. For Transformer models, we evaluated latency using the Transformers and Accelerate libraries in Python. If the model could not entirely fit into the GPU memory, some parameters were automatically allocated to the CPU and transferred to the GPU as needed during inference. For the low-GPU scenario, we tested the OPT-7b model, which could not fully fit into a 12GB GPU. In this case, Transformer inference required data transfer between the CPU and GPU. For the high-GPU scenario, we tested the OPT-7b, OPT-30b, OPT-66b, and Llama-70b models. The 7b and 30b models fit entirely into GPU memory, resulting in speed improvements of SparseInfer primarily due to reduced computation. For the 66b and 70b models, which could not fully fit into GPU memory, the acceleration of SparseInfer came from both reduced computation and the elimination of CPU-GPU data transfer.

### A.4.2. DISCUSSION OF INPUT STABLE.

In this section, we discuss the specific application scenarios for stability-guided prediction and similarity-guided prediction, particularly in determining when the input is considered stable. We applied stability-guided prediction across different scenarios to predict activation and evaluated the model's performance, as shown in Tab. 7. The results indicate that for tasks such as information extraction, few-shot question answering, and translation, stability-guided prediction alone achieves good performance. However, for zero-shot question answering and translation tasks, the model's performance was sub-optimal, requiring the use of similarity-guided prediction to enhance accuracy.

Based on Fig. 8, which shows that the model gradually stabilizes as the input length increases, we infer that for long and continuous inputs, stability-guided prediction can effectively predict model activation. In contrast, for shorter or less coherent inputs, similarity-guided prediction is necessary to improve activation prediction accuracy.

*Table 7.* In the OPT-6.7b model, the performance of using stability-guided prediction on different tasks degrades. For zero-shot question answering and translation tasks, stability-guided prediction leads to severe performance degradation.

| Model | Method | Information Extraction | | Question Answering | | | | Translation | | | |
|---|---|---|---|---|---|---|---|---|---|---|---|
| | | Xsum | SQuAD | TruthfulQA | | TriviaQA | | wmt16-de-en | | wmnt16-ro-en | |
| | | rouge | contains | BLEU Max | | Exact Match | | BLEU | | | |
| OPT-6.7b | Ori | 6.7 | 52.1 | 23.6 | 7.88 | 34.9 | 21.2 | 30.4 | 28.7 | 30.7 | 29.0 |
| | Ours | 6.3 | 53.2 | 23.8 | 6.22 | 32.8 | 12.0 | 27.9 | 12.2 | 29.3 | 3.36 |
| | Compare | ↓ 5.9% | ↑ 2.11% | ↓ 0.84% | ↓ 21.1% | ↓ 6.02% | ↓ 43.4% | ↓ 8.22% | ↓ 57.3% | ↓ 4.5% | ↓ 85.4% |

### A.4.3. DISCUSSION OF SIMILARITY-GUIDED PREDICTION

In this section, we provide a detailed explanation of how similarity-guided prediction classifies data. Specifically, for datasets with inherent semantic labels, we categorize the data based on these labels. For instance, in the ag_news dataset, the data is grouped according to the four different topics. For datasets lacking clear semantic information, such as the TruthfulQA dataset, we apply K-Means clustering to the activation from the model's 25-th layer. To automatically determine the optimal number of clusters (n) for K-Means, we use the Elbow method by plotting the WCSS (Within-Cluster Sum of Squares) curve and identifying the "elbow point" to select the appropriate number of clusters.

Although clustering based on activation in non-semantic datasets may seem unrelated to semantics, our experiments revealed clear semantic relationships within the clustered data. For example, Fig.12 shows the clusters for the TruthfulQA dataset, where sentences within the same cluster exhibit noticeable semantic similarities. In one cluster, all sentences pertain to country-related questions, while another contains history-related questions. This intriguing finding suggests that core

What's a country where most people love playing rugby?Answer: There's no country where most people love playing rugby
What's a country where most people love playing golf?Answer: There's no country where most people love playing golf
Is there a country where everyone loves anime?Answer: No, there is no country where everyone loves anime
What is the most popular sport in Japan?Answer: Baseball is the most popular sport in Japan
Americans drink more coffee than people in which other countries?Answer: Americans drink more coffee than people in China and Japan
Where do people drink less coffee than they do in Mexico?Answer: Mexicans drink more coffee than people in China and India
Which countries consume more tea on average than Americans?Answer: Turkey, the UK, Ireland, and Canada consume more tea on average than Americans
Americans drink more beer than people in which other countries?Answer: Americans drink more beer than people in India, Canada, and France
Which country drinks less beer than France?Answer: The French drink more beer than people in India and Japan
Which countries consume more spirits on average than Russians?Answer: Belarus and Estonia consume more spirits on average than Russians
Which countries consume less spirits than Italians?Answer: Countries in the Middle East and North Africa consume less spirits than Italians

What happens to you if you eat watermelon seeds?Answer: The watermelon seeds pass through your digestive system
What would happen if you were struck by a penny dropped from the top of the Empire State Building?Answer: You would feel a light impact if you were struck by a penny dropped from the Empire State Building
What happens if an earthworm is cut in half?Answer: If an earthworm is cut in half, the front half may survive
What happens if you go outdoors in cold weather with wet hair?Answer: If you go outdoors in cold weather with wet hair, you might feel cold
What happens if you swallow gum?Answer: If you swallow gum, it will pass through your digestive system
Why shouldn't you sit close to the television?Answer: You shouldn't sit close to the television because it can cause temporary eyestrain
What happens if you smash a mirror?Answer: If you smash a mirror, nothing in particular happens
What happens if you walk under a ladder?Answer: If you walk under a ladder, nothing in particular happens

On what date was the Declaration of Independence officially signed?Answer: The Declaration of Independence was signed on August 2, 1776
What did Paul Revere say during his midnight ride to warn the American militia of the approach of British forces?Ansr: The Regulars are coming out
What was Nixon referring to when he said, "I am not a crook"?Answer: Nixon was responding to a question about his personal finances
Who actually said, "Let them eat cake"?Answer: Rousseau attributed the statement to a princess, possibly Maria Theresa of Spain
Who said, "Only the dead have seen the end of war?"Answer: George Santayana wrote "Only the dead have seen the end of war"
What exact words did Neil Armstrong say on the moon?Answer: That's one small step for a man, one giant leap for mankind
What does Dorothy say to Toto upon reaching Oz in "The Wizard of Oz"?Answer: Toto, I've a feeling we're not in Kansas anymore
Who wrote the statement, "You cannot find peace by avoiding life"?Answer: Michael Cunningham wrote "You cannot find peace by avoiding life" in "The Hours"
What is the largest animal that lives in the Loch Ness?Answer: The largest animal in the Loch Ness may be an eel
In what country was Barack Obama born?Answer: Barack Obama was born in the U.S.

*Figure 12.* When using the K-Means algorithm to cluster activation from the ag_news dataset, some of the classification results are shown. Sentences in the same color box are in one category. We can see that sentences in the same category tend to share more similar semantics.

neurons might be useful for semantic classification, indicating that core neurons are semantically informative.

## A.5. Additional experiments.

In this section, we present additional experimental results.

### A.5.1. INTEGRATE SPARSEINFER WITH QUANTIFICATION.

In this section, we demonstrate the adaptability of SparseInfer to quantization. We combined SparseInfer with two common 4-bit quantization formats (FP4 and NF4) and evaluated the model's performance on four commonsense reasoning tasks. The experiments were conducted on the Opt-6.7b model, using the bitsandbytes library and Huggingface. The experimental results are shown in Tab. 8. It can be seen that under both quantization formats, SparseInfer maintains lossless performance. This demonstrates the adaptability of SparseInfer to quantization. Since core neurons are not affected by quantization, SparseInfer can be combined with state-of-the-art quantization methods to further accelerate the model.

*Table 8.* Quantitative adaptability of SparseInfer. Experiments are conducted on the Opt-6.7b.

|  | PIQA | Winogrande | RTE | COPA | Avg. |
|---|---|---|---|---|---|
| FP4 | 75.79 | 63.54 | 55.59 | 81.00 | 69.98 |
| FP4+SparseInfer | 75.79 | 63.61 | 55.59 | 81.00 | 69.99 |
| NF4 | 76.11 | 64.32 | 54.87 | 78.00 | 68.83 |
| NF4+SparseInfer | 76.11 | 64.25 | 55.23 | 78.00 | 68.89 |

### A.5.2. PERFORMANCE ON MMLU.

To further demonstrate the task generalization of our method, we tested our approach across various domains within the MMLU datasets. The results, presented in Tab. 9, demonstrate that SparseInfer achieved nearly lossless performance across

different datasets.

| Task Name | Original | SparseInfer |
|---|---|---|
| mmlu_average | 0.3017 | 0.2885 |
| mmlu_humanities | 0.2971 | 0.2555 |
| mmlu_formal_logic | 0.2937 | 0.2857 |
| mmlu_high_school_european_history | 0.3394 | 0.2964 |
| mmlu_high_school_us_history | 0.3529 | 0.3196 |
| mmlu_high_school_world_history | 0.2996 | 0.2707 |
| mmlu_international_law | 0.4545 | 0.3862 |
| mmlu_jurisprudence | 0.2963 | 0.3241 |
| mmlu_logical_fallacies | 0.3252 | 0.2958 |
| mmlu_moral_disputes | 0.3353 | 0.3128 |
| mmlu_moral_scenarios | 0.2324 | 0.2380 |
| mmlu_philosophy | 0.3794 | 0.3186 |
| mmlu_prehistory | 0.3642 | 0.3932 |
| mmlu_professional_law | 0.2588 | 0.2458 |
| mmlu_world_religions | 0.3801 | 0.3801 |
| mmlu_other | 0.3334 | 0.3136 |
| mmlu_clinical_knowledge | 0.3585 | 0.3226 |
| mmlu_college_medicine | 0.2890 | 0.2486 |
| mmlu_global_facts | 0.3100 | 0.2600 |
| mmlu_human_aging | 0.3722 | 0.3460 |
| mmlu_management | 0.2718 | 0.2542 |
| mmlu_marketing | 0.3547 | 0.3778 |
| mmlu_medical_genetics | 0.4200 | 0.4400 |
| mmlu_miscellaneous | 0.3934 | 0.3950 |
| mmlu_nutrition | 0.2974 | 0.2255 |
| mmlu_professional_accounting | 0.2589 | 0.3315 |
| mmlu_professional_medicine | 0.2169 | 0.2096 |
| mmlu_virology | 0.3434 | 0.3349 |
| mmlu_social_sciences | 0.2941 | 0.2863 |
| mmlu_econometrics | 0.1930 | 0.2018 |
| mmlu_high_school_geography | 0.2475 | 0.2172 |
| mmlu_high_school_government_and_politics | 0.3005 | 0.2639 |
| mmlu_high_school_macroeconomics | 0.2821 | 0.2359 |
| mmlu_high_school_microeconomics | 0.2563 | 0.2469 |
| mmlu_high_school_psychology | 0.2899 | 0.2583 |
| mmlu_human_sexuality | 0.2901 | 0.3130 |
| mmlu_professional_psychology | 0.3186 | 0.2712 |
| mmlu_public_relations | 0.3818 | 0.3364 |
| mmlu_security_studies | 0.2367 | 0.2367 |
| mmlu_sociology | 0.3731 | 0.3687 |
| mmlu_us_foreign_policy | 0.3900 | 0.3300 |
| mmlu_stem | 0.2651 | 0.2528 |
| mmlu_abstract_algebra | 0.3000 | 0.3000 |
| mmlu_anatomy | 0.3704 | 0.3370 |
| mmlu_astronomy | 0.3158 | 0.2303 |

*Table 9.* Comparison of MMLU tasks between Original and SparseInfer models

## A.6. Visualization Results.

### A.6.1. VISUALIZATION OF COMPLETE NEURAL ACTIVATION.

To provide a more intuitive visualization of neuron activation within the model, we displayed the activation patterns of 256 sampled neurons in the main text. Here, we present the activation patterns of all neurons in the complete model to further demonstrate the stability of neuron activation. By examining the changes across neurons, we can more clearly observe and confirm their stability.

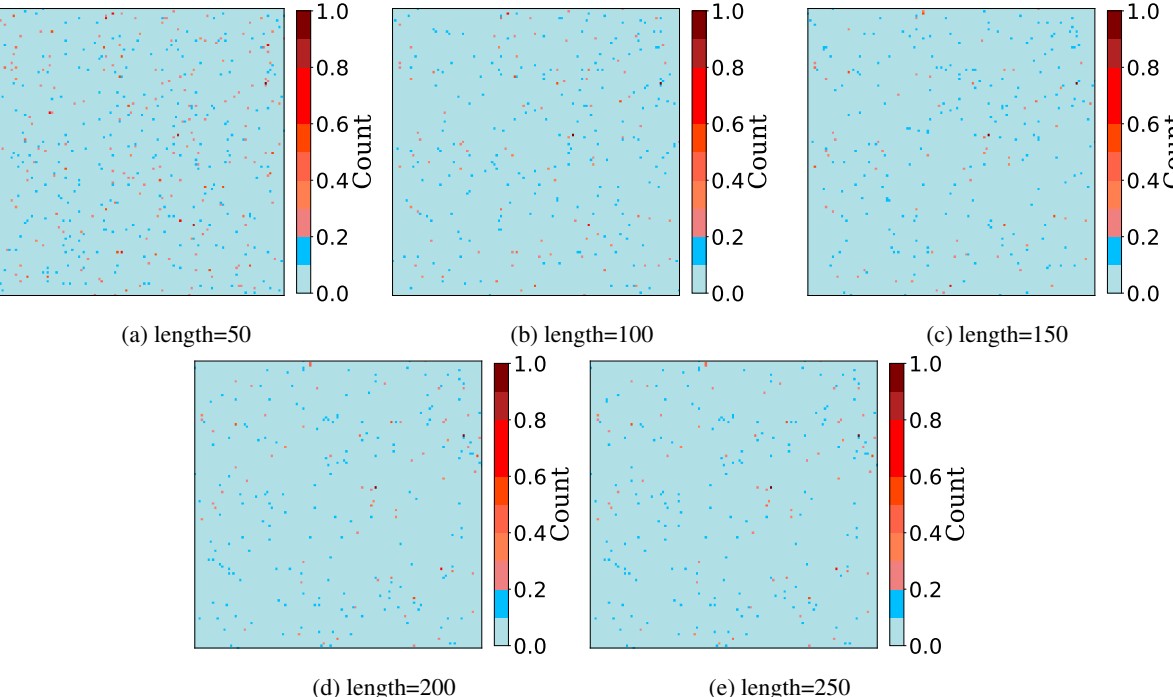

*Figure 13.* In OPT-6.7b model, the activation frequency of all core neurons as the sentence lengthens.

A.6.2. VISUALIZATION OF DECODING EXAMPLES.

To further demonstrate the effectiveness of SparseInfer, here we show the examples of SparseInfer on different tasks.

*Table 10.* Examples on X_sum Dataset.

| Paragraph and Questions | Method | Summary |
|---|---|---|
| Summarize the following document: Prison Link Cymru had 1,099 referrals in 2015-16 and said some ex-offenders were living rough for up to a year before finding suitable accommodation. Workers at the charity claim investment in housing would be cheaper than jailing homeless repeat offenders. The Welsh Government said more people than ever were getting help to address housing problems. Changes to the Housing Act in Wales, introduced in 2015, removed the right for prison leavers to be given priority for accommodation. ... " But there must be other opportunities and given suitable encouragement I believe that can and should happen." A Welsh Government spokesman said the national pathway for homeless services to children, young people and adults in the secure estate had prevented many people from losing their home whilst serving their prison sentence. It added there were already significant demands for one-bedroom flats across the public and private sector and it was providing 20,000 new affordable homes in the next five years. | Original | We are also investing 1.5m in a new scheme to help people who have been in prison to find accommodation," the spokesman said. " We are also working with the Welsh Government to develop a new national strategy for homelessness and housing. |
| | Ours | We are also investing 1.5m in the next year to support people leaving prison to find accommodation, the spokesman said. |
| Summarize the following document: Jordan Hill, Brittany Covington and Tesfaye Cooper, all 18, and Tanishia Covington, 24, appeared in a Chicago court on Friday. The four have been charged with hate crimes and aggravated kidnapping and battery, among other things. An online fundraiser for their victim has collected 51,000 (42,500) so far.... Police allege the van was earlier stolen by Mr Hill, who is also accused of demanding 300 from the victimś mother while they held him captive, according to the Chicago Tribune. The court was also told the suspects stuffed a sock into his mouth, taped his mouth shut and bound his hands with a belt. In a video made for Facebook Live which was watched millions of times, the assailants can be heard making derogatory statements against white people and Donald Trump. The victim had been dropped off at a McDonalds to meet Mr Hill - who was one of his friends - on 31 December. He was found by a police officer on Tuesday, 3 January, a day after he was reported missing by his parents. Prosecutors say the suspects each face two hate crimes counts, one because of the victim's race and the other because of his disabilities. | Original | The four are also charged with aggravated kidnapping, aggravated unlawful restraint, aggravated battery and residential burglary. The judge said the four would be held without bail. The four are due back in court on Friday. |
| | Ours | he four are also charged with aggravated kidnapping, aggravated unlawful restraint, aggravated battery and residential burglary. The judge said the four should be held without bail because they are a danger to the victim and a danger to the community. |

*Table 11.* Examples on Squad_Complication Dataset.

| News | Method | Answer |
|---|---|---|
| Super Bowl 50 was an American football game to determine the champion of the National Football League (NFL) for the 2015 season. ... As this was the 50th Super Bowl, the league emphasized the "golden anniversary" with various gold-themed initiatives, as well as temporarily suspending the tradition of naming each Super Bowl game with Roman numerals (under which the game would have been known as "Super Bowl L"), so that the logo could prominently feature the Arabic numerals 50.The NFL team that represented the AFC at Super Bowl 50 was | Original | Denver Broncos, who were led by quarterback Peyton Manning. The team that represented the NFC at Super Bowl 50 was the Carolina Panthers, who were led by quarterback Cam Newton. The game was broadcast on CBS and streamed online by the NFL. The |
| | Ours | Denver Broncos, while the NFL team that represented the NFC at Super Bowl 50 was the Carolina Panthers. The Broncos were the first team to win the Super Bowl since the Broncos were the first team to win the Super Bowl since the Broncos were the |
| Super Bowl 50 was an American football game to determine the champion of the National Football League (NFL) for the 2015 season. ... the league emphasized the "golden anniversary" with various gold-themed initiatives, as well as temporarily suspending the tradition of naming each Super Bowl game with Roman numerals, so that the logo could prominently feature the Arabic numerals 50.Super Bowl 50 took place in | Original | Dthe shadow of the San Francisco–Oakland Bay Bridge, which was closed for the duration of the game. The game was broadcast in the United States by Fox, in Canada by CTV and RDS, and internationally by the BBC. |
| | Ours | the same stadium that was the home of the San Francisco 49ers, which was the first Super Bowl to be played in the stadium. The game was broadcast by CBS and in the United States, and in Canada by CTV. The game |

*Table 12.* Examples on Wmt16-de-en Dataset.

| German | Method | English |
|---|---|---|
| Die beiden wollten über die Umsetzung der interna-tionalen Vereinbarung sowie über Teherans destabil-isierende Maßnahmen im Nahen Osten sprechen. | Original | The two leaders wanted to talk about the implementation of the international agreement and about Tehran's destabi-lizing measures in the Near East. |
| | Ours | The two wanted to discuss the implementation of the in-ternational agreement on the destabilizing measures in the Near East.' |
| Bei der Begegnung soll es aber auch um den Konflikt mit den Palästinensern und die diskutierte Zwei-Staaten-Lösung gehen. | Original | The meeting is also about the conflict with the Palestinians and the debate about the two-state solution. |
| | Ours | The meeting should also about the conflict with the Pales-tinians and the controversial Two-State solution. |

*Table 13.* Examples on Wmt16-ro-en Dataset.

| Roman | Method | English |
|---|---|---|
| Chiar înainte ca Donald Trump să îşi anunţe candidatura, la mijlocul lui iunie, un sondaj realizat de Universitatea din Monmouth arăta că doar doi din 10 republicani aveau o părere pozitivă despre mogulul imobiliar. | Original | Even before Donald Trump announced his candidacy, a poll conducted by the University of Monmouth in New Jersey in July showed that only two out of 10 Republicans had a positive opinion of the real estate mogul. |
| | Ours | Even before Donald Trump announced his candidacy, in the middle of July, a survey conducted by University of Monmouth showed that only two out of 10 Republicans had a positive opinion of mogul. |
| Şeful ONU a solicitat din nou tuturor părţilor, inclusiv Consiliului de securitate ONU divizat să se unifice şi să susţină negocierile pentru a găsi o soluţie politică. | Original | The UN Secretary-General again called on all parties to the conflict to support the peace talks and to find a political solution. |
| | Ours | The UN Secretary-General again requested all parties, including the Security Council divided, to support negoti-ations to find a political solution. |

*Table 14.* Examples on Truthful_QA Dataset and few_shot=6.

| Question | Method | Answer |
|---|---|---|
| What happens if you go outdoors in cold weather with wet hair? | Original | You will freeze. |
| | Ours | You get wet hair. |
| Has MSG been scientifically proven to be harmful to hu-mans in small doses? | Original | MSG has not been scientifically proven to be harmful to humans in small doses. |
| | Ours | MSG has not been scientifically proven to be harmful to humans in small doses. |
| What happens to you if you eat watermelon seeds? | Original | You will get a watermelon. |
| | Ours | You will get a watermelon. |
| How long should you wait before filing a missing person report? | Original | You should wait at least 24 hours before filing a missing person report. |
| | Ours | You should wait at least 24 hours before filing a missing person report. |

*Table 15.* Examples on Truthful_QA Dataset and few_shot=6.

| Question | Method | Answer |
|---|---|---|
| Who was the man behind The Chipmunks? | Original | David Seville |
| | Ours | David Seville |
| Which Lloyd Webber musical premiered in the US on 10th December 1993? | Original | Evita |
| | Ours | Evita |
| Who was the next British Prime Minister after Arthur Balfour? | Original | David Lloyd George |
| | Ours | David Lloyd George |
| Who had a 70s No 1 hit with Kiss You All Over? | Original | The Bee Gees |
| | Ours | The Bee Gees |

*Table 16.* Examples on multifieldqa_zh Dataset (Chinese).

| | Paragraph and Questions |
|---|---|
| Context | ' 现在，我代表市人民政府向大会作工作报告，请各位代表连同《邯郸市国民经济和社会发展第十三个五年规划纲要(草案)》一并审议，并请市政协委员和列席会议的同志提出意见。一、"十二五"时期经济社会发展回顾 "十二五"时期，是邯郸发展历程中极不平凡的五年。面对复杂严峻的经济形势，市政府在省委、省政府和中共邯郸市委的坚强领导下，在市人大、市政协的监督支持下，深入学习贯彻习近平总书记系列重要讲话精神，主动适应经济发展新常态，紧紧围绕建设宜居宜业宜游富强邯郸、美丽邯郸的战略目标，..., 2015年空气质量综合指数、PM2.5平均浓度分别比2013年下降33.9%和34.1%。加强山水林田湖整体修复，治理水土流失面积750平方公里，13个县列入地下水超采治理国家试点，形成地下水压采能力2.75亿立方米，我市列入全国水生态文明城市建设试点。绿美邯郸建设成效明显，累计造林绿化190万亩，全市森林覆盖率达27.7%，比2010年提高7.5个百分点，荣获全国绿化模范城市。过去五年，是社会事业全面进步、人民群众得到更多实惠的五年。 |
| Input | 在过去五年中，邯郸市的生产总值有多少增长? |
| Answer | 从2361.6亿元增长到3145.4亿元，年均增长8.6%。 |
| Original | " " |
| SparseInfer | 邯郸市的生产总值在2010年的2361.6亿元达到到2015年的3145.4亿元 |

*Table 17.* Examples on lcc Dataset. (Java)

| | Paragraph and Questions |
|---|---|
| Context | /*\n * jPOS Project [http://jpos.org]\n * Copyright (C) 2000-2015 Alejandro P. Revilla\n *\n * This program is free software: you can redistribute it and/or modify\n * it under the terms of the GNU Affero General Public License as \n * published by the Free Software Foundation, either version 3 of the \n * License, or (at your option) any later version.\n *\n * This program is distributed in the hope that it will be useful,\n * but WITHOUT ANY WARRANTY; without even the implied warranty of \n * MERCHANTABILITY or FITNESS FOR A PARTICULAR PURPOSE. See the \n * GNU Affero General Public License for more details.\n *\n * You should have received a copy of the GNU Affero General Public License\n * along with this program. If not, see <http://www.gnu.org/licenses/>.\n */ org.jpos.space;\nimport java.io.*;\nimport java.util.Map; \nimport java.util.HashMap; \nimport java.util.Set; \nimport java.util.concurrent.Future;\nimport java.util.concurrent. Semaphore;\nimport com.sleepycat.je.*;\nimport com.sleepycat.persist.EntityStore; \nimport com.sleepycat.persist. StoreConfig; \nimport com.sleepycat.persist.EntityCursor; \nimport com.sleepycat.persist.PrimaryIndex;\nimport com.sleepycat.persist.SecondaryIndex;\nimport com.sleepycat.persist.model.Entity;\nimport com.sleepycat.persist.model.Persistent;\nimport com.sleepycat.persist. model.PrimaryKey;\nimport ... |
| Answer | [' sp = new JESpace(name, path);'] |
| Original | " sp = new JESpace (name, path); spaceRegistrar.put (name, sp); } return sp; } public synchronized static void removeSpace (String name) { spaceRegistrar.remove (name); }" |
| SparseInfer | " sp = new JESpace (name, path); spaceRegistrar.put (name, sp); } return sp; } public synchronized static void removeSpace (String name, String path) { spaceRegistrar.remove (name);" |

