# OpenReview forum: "SparseInfer: Accelerating Large Language Model Inference with Semantics-Inspired Adaptive Sparse Activation"
_ICML.cc/2026/Conference — ICML 2026 regular_

### Official Review · Reviewer_xfGH · 2026-02-26

**Soundness:** 2
**Presentation:** 3
**Significance:** 3
**Originality:** 3
**Overall Recommendation:** 5
**Confidence:** 4

**Summary:**

The paper assesses the key issue of high computational and memory costs during Large Language Model (LLM) inference, proposing a novel solution to accelerate deployment on resource-constrained devices. The authors present a general area of adaptive sparse activation, introducing **SparseInfer**, a training-free framework that shifts from token-level to sentence-level neuron selection. By defining "core neurons" essential for processing a specific sentence, the method leverages observed stability and semantic similarity in activation patterns to predict and fix the active neuron subset during the pre-filling stage. This approach eliminates the need for auxiliary MLP predictors used in prior works (e.g., PowerInfer, DejaVu), thereby reducing overhead. Experimental results on OPT and LLaMA families demonstrate significant speedups (up to 10.33×) with negligible performance loss across various tasks, including long-context generation and multi-turn dialogue.

**Compliance With Llm Reviewing Policy:**

Affirmed.

**Final Justification:**

This paper effectively explores how to further optimize activation sparsity, demonstrating promising results in the post-training scenario with good generalizability. The author’s responses have also adequately addressed my concerns, and therefore I am willing to raise my score.

**Key Questions For Authors:**

1.  **Generalization to Other Architectures:** The experiments focus exclusively on OPT and LLaMA models. Can the authors demonstrate that the "stability" and "similarity" insights hold for other modern architectures like **Mistral** or **Qwen**? If not, what are the potential limitations of applying SparseInfer to these models?
2.  **Detailed Workflow for Similarity-Guided Prediction:** The paper mentions using clustering for similarity-guided prediction but lacks a concrete algorithmic description. Could the authors provide a step-by-step breakdown of how an input sentence is mapped to a cluster during inference? What is the computational cost of this mapping, and does it negate the latency benefits gained from sparse activation?
3.  **Layer-wise Hyperparameter Optimization:** The current approach applies uniform hyperparameters ($\alpha, \beta, \gamma$) across all layers. Given that earlier and deeper layers often capture different types of information, have the authors explored layer-specific tuning? Would adaptive thresholds per layer yield better accuracy-efficiency trade-offs?
4.  **Compatibility with MoE Models:** With the rising prevalence of native Mixture-of-Experts (MoE) models, how does SparseInfer interact with existing MoE architectures? Is there a conflict between the proposed "core neuron" selection and the native router mechanisms in MoE models, or can this method be applied to further sparsify experts within an MoE layer?

**Limitations:**

The authors should explicitly discuss the following limitations in the final version:
1.  **Architectural Generalizability:** The lack of evaluation on non-LLaMA/OPT models (e.g., Mistral, Qwen, Gemma) limits the confidence in the method's universal applicability.
2.  **Hyperparameter Robustness:** The reliance on empirically fixed hyperparameters without extensive sensitivity analysis or layer-wise adaptation strategies may hinder optimal performance across diverse tasks and models.
3.  **MoE Compatibility:** The paper does not address whether this dense-to-sparse conversion strategy is applicable or beneficial for already-sparse MoE models, which is a significant gap given the current trend in LLM development.

**Strengths And Weaknesses:**

**Strengths:**

*   **Significance (Good):** The work effectively addresses the performance bottlenecks of existing MLP-based prediction strategies in activation sparsity. By removing the predictor overhead and shifting to sentence-level granularity, it offers a practical and efficient solution that achieves superior performance-speed trade-offs compared to state-of-the-art baselines.
*   **Originality (Good):** The paper provides valuable insights into the semantic correlation of activation sparsity, specifically the observation that core neurons exhibit stability and similarity relative to sentence semantics. This theoretical grounding distinguishes it from purely heuristic pruning methods.
*   **Presentation (Good):** The manuscript is well-written and clearly structured. The motivation is compelling, and the logical flow from definition to exploration and finally to the proposed framework is easy to follow.

**Weaknesses:**

*   **Soundness:** While the experimental evaluation is extensive, there are concerns regarding generalizability and implementation clarity:
    1.  **Limited Model Diversity:** The evaluation is restricted to OPT and LLaMA family models (ReLU and SiLU activations). It remains unclear whether the observed stability and similarity properties hold for other prominent architectures like **Mistral** or **Qwen**, which employ different structural designs (e.g., Grouped Query Attention, sliding windows). Without validation on these models, the claim of broad generalizability is weakened.
    2.  **Unclear Implementation of Similarity-Guided Prediction:** The description of the **similarity-guided prediction** workflow (Section 3.1) lacks sufficient detail. Specifically, the process of clustering the training dataset, determining the optimal number of clusters, and mapping an incoming unstable input to a specific cluster during inference is not rigorously defined. The reliance on K-Means (mentioned in Appendix A.4.3) raises questions about latency overhead during the pre-filling stage for unseen data distributions.
    3.  **Hyperparameter Sensitivity:** The method relies on hyperparameters $\alpha$, $\beta$, and $\gamma$. The current study uses empirically determined values (e.g., 0.2 or 0.4) based on limited ablation. There is no exploration of whether these parameters should vary across different layers (e.g., early vs. deep layers) or if a uniform setting is truly optimal. The sensitivity of performance to these thresholds needs deeper analysis.
*   **Minor Issues:** There are several typographical errors, such as missing spaces in Figure 1 captions (line 254) and inconsistent unit formatting (e.g., "Hardware" section should explicitly use "GB").

---

> ### Author Rebuttal · Authors · 2026-03-30
>
> We sincerely thank the reviewer for the time and effort spent reviewing our paper. **During the rebuttal period, we conducted many additional experiments to further strengthen our paper. Please refer to https://drive.google.com/file/d/1mzeIdgzXQ4tnFyT39dPNJ9yfYF_cMJdn/view?usp=sharing**
>
> ## 1. Generalization to Other Architectures
>
> We thank the reviewer for this suggestion. We have verified SparseInfer on Qwen3-4B-Base, Qwen3-8B-Base, Mistral-Nemo-Base, and Mistral-7B-v0.3, confirming its effectiveness on modern non-ReLU architectures (**see Figs. 1 & 2 of additional experiments**).
>
> - **Adaptation to SwiGLU:** SparseInfer reduces high-dimensional FFN computation by focusing on the intermediate activation $|z|$, where $z = \text{SiLU}(xW_g) \odot (xW_u)$. Since $z$ directly determines the output $y = zW_d$, we achieve sparse inference by retaining only the columns/rows corresponding to the largest entries in $|z|$.
> - **Sparsity Verification:** Fig. 1 shows that $|z|$ is highly sparse across all four models, with a small subset of "core neurons" dominating the total activation.
> - **Performance:** Following our evaluation protocol, Fig. 2 shows that retaining only **40%** of static sentence-level core neurons results in **negligible PPL increases**.
>
> These results demonstrate that our insights on core neurons and the SparseInfer mechanism generalize effectively to modern SwiGLU-based LLMs.
>
> ---
>
> ## 2. Detailed Workflow for Similarity-Guided Prediction
>
> We thank the reviewer for pointing out that our description of the similarity-guided prediction method was not detailed enough. In the revised version, we will add a step-by-step description of the mapping process during inference.
>
> **Offline Preparation:** We first cluster the training corpus into $K$ semantic groups using K-Means on sentence embeddings. For each cluster and layer, we identify the core neurons of all member sentences and aggregate their activation frequencies. Finally, we pre-compute a fixed "cluster-level neuron prototype" by selecting the Top-$\gamma$ most frequently activated neurons in each group.
>
> **Inference-stage workflow.** The inference stage consists of the following steps:
>
> 1. Given an input sentence $x$, we first compute its semantic representation $e(x)$ during prefilling.
>
> 2. We then assign $x$ to its nearest semantic cluster by
>    $ k^* = \arg\min_k \|e(x)-\mu_k\|_2^2. $
>
> 3. Once the target cluster index $ k^* $ is determined, we directly retrieve the pre-computed cluster-level neuron prototypes $\{M_{k^*,i}\}$.
>
> 4. These cluster-level sparse neuron sets are then used as the predicted active neurons during decoding.
>
> In summary, K-Means is only used offline to build semantic groups. During inference, we only perform a lightweight pipeline: **embedding computation + nearest-centroid matching + table lookup**.
>
> **Computational overhead.** This similarity-guided mapping requires only one sentence-embedding computation and one nearest-centroid search in a low-dimensional space. In practice, this step usually takes only milliseconds on a GPU, so its overhead is negligible compared with the end-to-end latency of LLM generation.
>
> **Adaptive online K-Means update.** We also developed a new online K-Means mechanism, as described in **our response to Reviewer 6bKQ, Question 2**. This mechanism allows the system to continuously detect out-of-distribution inputs during inference and dynamically update both the centroids and the number of clusters, making it better suited for real-world deployment.
>
> ---
>
> ## 3. Layer-wise Hyperparameter Optimization
>
> We thank the reviewer for this suggestion. To investigate layer-wise hyperparameters, we compared our **fixed-ratio** strategy with a **magnitude-based** approach (retaining neurons until they reach a fixed percentage of total layer activation) on Qwen3-4B-Base.
>
> Results (**Fig. 5 of additional experiments**) show that while layer-wise tuning slightly improves the accuracy-efficiency trade-off, the gains are marginal. Because most middle layers exhibit similar activation distributions, a uniform ratio remains a robust and effective default.
>
>
>
> ---
>
> ## 4. Compatibility with MoE Models
>
> We thank the reviewer for this suggestion. We verified SparseInfer on **Qwen1.5-MoE-A2.7B** and **Mixtral-8x7B-v0.1** by treating the intermediate dimensions of all experts as a unified neuron pool.
>
> Key Findings:
>
> - **Orthogonality:** SparseInfer is complementary to the MoE router. While the router selects **which experts** to activate, SparseInfer selects the most critical **neurons within those experts**.
> - **Performance (Fig. 6 of additional experiments):** MoE models exhibit the same sparsity pattern as dense models, maintaining high performance even with a small fraction of core neurons.
>
> This confirms that SparseInfer acts as an additional, orthogonal layer of sparsity that further optimizes MoE expert internal efficiency.
>
> **We will include these results in the revised manuscript.**

---

> > ### Author Rebuttal · Reviewer_xfGH · 2026-04-02
> >
> > Thank you for the detailed and thoughtful response. I have reviewed your rebuttal, and it has fully addressed all of my concerns. I have no further questions at this time.

---

> > > ### Author Response · Authors · 2026-04-02
> > >
> > > Dear Reviewer,
> > >
> > > Thank you very much for your updated evaluation of our paper. We are truly grateful that you feel we have addressed your concerns, and we sincerely appreciate your decision to raise the score.
> > >
> > > We also deeply appreciate the time and effort you devoted to reviewing our paper. Thank you again for your thoughtful comments and valuable suggestions.
> > >
> > > Best regards,
> > >
> > > The Authors

---

### Official Review · Reviewer_6bKQ · 2026-03-08

**Soundness:** 2
**Presentation:** 3
**Significance:** 2
**Originality:** 2
**Overall Recommendation:** 4
**Confidence:** 4

**Summary:**

The paper addresses the key issue of how to accelerate large language models (LLMs) inference while maintaining performance, focusing on activation sparsity as an alternative to model compression. The authors present a general area of adaptive sparse inference and propose SparseInfer, a sentence-level sparse activation method that identifies a set of “core neurons” sufficient for processing a given sentence. By exploiting the observed stability and semantic similarity of these core neurons, the method predicts a fixed neuron subset during the pre-filling stage and reuses it during decoding, avoiding per-token prediction and additional MLP predictors. Experiments across multiple models and tasks show that SparseInfer maintains comparable accuracy while achieving up to 10.33x inference speedup and reduced memory usage compared to prior sparse inference approaches.

**Compliance With Llm Reviewing Policy:**

Affirmed.

**Final Justification:**

The rebuttal addresses several concerns, but many improvements rely on additional mechanisms (SparseInfer++) beyond the original method, suggesting some issues were not fully considered initially. Some aspects (e.g., sentence length and clustering) remain unclear.

Overall, my concerns are partially addressed, and I slightly increase my score while maintaining a cautious assessment.

**Key Questions For Authors:**

1.	The proposed method appears to be limited to ReLU-based sparsity, which may not generalize well to more recent LLM architectures. The authors primarily use OPT models for experimentation, which, while useful, are now somewhat outdated. To strengthen the paper, it would be beneficial to include more analysis on non-ReLU activations, such as SwiGLU, which are increasingly used in modern LLMs. Additionally, the experiments on the OPT series models are insufficient, as they do not reflect the current state of the field. It would be valuable to include experimental results on newer models like Qwen-3 [1], which represent more recent advancements in model architecture.

2.	The proposed method makes a strong assumption that the input sentence has stable semantics. However, when the input lacks semantic stability, the authors suggest using similarity-guided prediction, which involves clustering based on the training set. Yet, how should the method handle unseen domain data that were not part of the training set? How are the number of clusters determined, and how are they formed? More details on these aspects are needed. This limitation could restrict the method's applicability in real-world scenarios, where inputs may be highly diverse or unfamiliar, such as in reasoning tasks that require divergent chains of thought.

3.	Moreover, the method for determining the appropriate sentence length for the proposed approach remains unclear. For instance, how should this be determined for unknown tasks? Additionally, the ideal sentence length may vary across different task domains, which adds complexity to the method's application.

4.	In Section 2.2, the authors mention that the stability of core neurons is related to semantic stability, which is an intuitive observation. However, is there an alternative method to quantitatively measure this stability?

5.	Most of the experiments focus on relatively simple tasks. How does the proposed method perform on more complex tasks, such as HumanEval [2] and GSM8K [3]? How does the method generalize to these more challenging tasks?

Reference:

[1] Qwen3 Technical Report. arXiv 2025.

[2] Evaluating Large Language Models Trained on Code. arXiv 2021.

[3] Training Verifiers to Solve Math Word Problems. arXiv 2021.

**Limitations:**

The authors acknowledge the potential societal implications of their work but do not provide a detailed discussion of the limitations or potential negative impacts.

**Strengths And Weaknesses:**

Strengths:
The paper presents a novel approach to accelerating large language model (LLM) inference by introducing SparseInfer, a method that utilizes sentence-level sparse activation rather than the typical token-level approach. This design helps to significantly reduce the computational cost associated with LLMs, especially by avoiding the overhead of MLP-based activation predictors used in previous sparse activation methods.

Weaknesses:
The main weaknesses of the paper include its limited generalization to newer LLM architectures, as the method primarily focuses on ReLU-based sparsity and uses outdated OPT models, which may not apply well to models with non-ReLU activations like SwiGLU. Additionally, the assumption that input semantics remain stable is problematic, as it is unclear how the method handles unseen domain data or inputs with fluctuating semantics, which limits its applicability in real-world scenarios. Furthermore, the method for determining the appropriate sentence length is not well defined, and the approach may vary across different tasks and domains, complicating its practical use in diverse settings.

---

> ### Author Rebuttal · Authors · 2026-03-30
>
> We sincerely thank the reviewer for the time and effort spent reviewing our paper. **During the rebuttal period, we conducted many additional experiments to further strengthen our paper. Please refer to https://drive.google.com/file/d/1mzeIdgzXQ4tnFyT39dPNJ9yfYF_cMJdn/view?usp=sharing**
>
> ---
> ## 1. Q1: Scalability to New Model Architectures
>
> We validated SparseInfer on four models. Results (**Figs. 1&2**) confirm SparseInfer's effectiveness across these modern architectures. For detailed analysis, please refer to our response to **Reviewer xfGH, Q1**.
>
> ---
> ## 2. Q2, Q3, Q5: How SparseInfer Can Be Applied Robustly in Real-World Settings
>
> To address concerns regarding real-world reliability—specifically strategy selection, complex tasks, and OOD handling—we developed **SparseInfer++**. While retaining the "core neuron" concept, it introduces three adaptive mechanisms to ensure robust practical deployment.
>
> ### **(1) Automatic switching between stability-guided and similarity-guided initialization**
>
> In the original SparseInfer, the choice between stability-guided and similarity-guided prediction is decided by the task setting. In SparseInfer++, this choice is made automatically.
>
> Let $P_t = C_{\alpha}^{\beta}(x_{1:t})$ denote the sentence-level core neurons computed from the first $t$ tokens $x_{1:t}$. We first measure the change between two consecutive tokens by
> $
> d_t = 1 - \frac{|P_t \cap P_{t-1}|}{|P_t \cup P_{t-1}|},
> $
> which is one minus the Jaccard overlap.
>
> We then smooth this sequence using EWMA:
> $
> S_t = \lambda d_t + (1-\lambda)S_{t-1}.
> $
> Here, $S_t$ measures how much the sentence-level core neurons change when a new token is added. If $S_t$ is small, the core neurons have become stable, and we use stability-guided initialization. If $S_t$ is large, we use similarity-guided initialization instead. In our experiments, we set $\lambda = 0.2$.
>
> ### **(2) Online update of core neurons during decoding**
>
> While original SparseInfer uses a fixed neuron set, SparseInfer++* introduces a dynamic update mechanism to handle complex reasoning tasks (e.g., math, coding) where activation patterns may shift.
>
> For layer $l$, let the current core-neuron set be $C^l$, and let $a_t^l(n)$ denote the activation of neuron $n \in C^l$ at decoding step $t$. We select the top-$k$ neurons in $C^l$ with the largest activations and denote this set by $T_t^l$. We then measure the change in the dominant activation pattern by
> $
> D_t^l = 1 - \frac{|T_t^l \cap T_{t-1}^l|}{|T_t^l \cup T_{t-1}^l|}.
> $
>
> $D_t^l$ reflects whether the current token significantly changes the activation pattern at step $t$. When $D_t^l > \tau_u$ exceeds a preset threshold, we treat this as a signal of semantic drift. In that case, we temporarily perform full activation at that layer and update the neuron scores using exponential smoothing:
> $
> m_t^l(n) = \rho |a_t^l(n)| + (1-\rho)m_{t-1}^l(n).
> $
>
> We then re-select the core-neuron set based on the updated scores. With this mechanism, SparseInfer++ keeps sparsity most of the time, while also being able to adapt to semantic changes during long decoding.
>
>  ### **(3) Online incremental clustering and OOD fallback for similarity-guided prediction**
>
> SparseInfer++ replaces offline clustering with an online incremental memory to handle evolving data distributions.
>
> Each cluster stores its centroid $\mu_k$, its size $N_k$, and layer-wise core-neuron statistics $H_k^l$. For a new input sample, we first extract its representation vector $z_x$ and find the nearest cluster:
> $ k^* = \arg\min_k \|z_x - \mu_k\|_2^2. $
>
> If the distance is higher than a preset threshold $\tau_{ood}$, we treat the input as an OOD sample and temporarily fall back to **full activation** during decoding. Once the input becomes stable enough to use stability-guided prediction, we record its stable core-neuron set and its representation vector.
> If the sample is far from all existing clusters, the system creates a new cluster. Otherwise, it incrementally updates the nearest cluster.
>
> ### **Overall pipeline of SparseInfer++**
>
> SparseInfer++ integrates three adaptive mechanisms to ensure real-world reliability:
>
> - **Prefilling:** Selects initial strategies based on neuron stability.
> - **Decoding:** Updates core neurons online to handle semantic shifts.
> - **Similarity-guided:** Updates clustering memory incrementally to learn from OOD inputs.
>
> The overall results are shown in **Table 1&2 of additional experiments**. We find that after adding the three mechanisms above, SparseInfer++ can handle these tasks while activating less than half fixed neurons, with only a small performance loss. We also provide validation results on OOD inputs in **Fig. 3**.
>
> ---
> ## 3. Q4: Other Ways to Measure Neuron Stability
>
> As described above, We use EWMA to compute $S_t$, measuring neuron stability by tracking changes per token. This metric enables adaptive strategy selection, and its effectiveness as a stability measure is confirmed by our experiments.

---

> > ### Author Rebuttal · Reviewer_6bKQ · 2026-04-04
> >
> > The authors have made a solid effort to address my concerns. However, many improvements rely on additional mechanisms introduced in SparseInfer++, which go beyond clarification and suggest that these issues were not fully considered in the original method.
> >
> > Some aspects (e.g., sentence length and clustering details) also remain unclear. Overall, my concerns are partially addressed, and I slightly increase my score.

---

> > > ### Author Response · Authors · 2026-04-06
> > >
> > > We sincerely thank the reviewer for recognizing that we have addressed part of their concerns and for raised the score. We are truly grateful for the reviewer’s valuable comments, which have helped make our paper stronger. At the same time, we would like to further clarify and explain several concerns that still remain.
> > >
> > > ---
> > > ## SparseInfer vs. SparseInfer++
> > >
> > > We acknowledge that SparseInfer++ was introduced as we more carefully considered situations that may arise in real-world deployment. However, SparseInfer++ does not depart from the core concepts and methodology proposed in SparseInfer:
> > >
> > > - **The mechanisms introduced in SparseInfer++ are all designed based on the properties of core neurons identified in SparseInfer.** For example, in the original paper, we proposed similarity-guided prediction because for relatively short inputs, the core neurons are not yet sufficiently stable. In SparseInfer++, we leverage exactly this insight: by measuring the stability of the input’s core neurons, we determine which prediction strategy should be used.
> > >
> > > - **The main additions in SparseInfer++ are engineering mechanisms intended to improve how SparseInfer can be deployed in complex real-world scenarios**, rather than introducing additional technical innovations. For instance, the online neuron update mechanism in SparseInfer++ is designed to handle long-text decoding in real-world applications, while dynamic k-means clustering is introduced to address situations such as encountering OOD samples. In summary, SparseInfer has already demonstrated the effectiveness of the proposed mechanism in relatively simple experimental settings, whereas SparseInfer++ further provides an effective deployment solution for dynamic real-world scenarios.
> > >
> > > In the revised paper, we will include SparseInfer++ as an additional section in the main text, so as to provide readers with a practical solution for deploying SparseInfer in challenging real-world environments.
> > >
> > > ---
> > > ## Clarification of Technical Details
> > >
> > > ### Regarding Input Sentence Length
> > >
> > > In SparseInfer++, the decision of prediction method is made automatically according to the stability of the input itself. Specifically, for a given input, we perform the following steps:
> > >
> > > 1. **Compute the current core neurons.** We extract the activation states of the first $t$ tokens $x_{1:t}$ and compute the current sentence-level core neurons, denoted as $P_t = C_{\alpha}^{\beta}(x_{1:t})$.
> > >
> > > 2. **Measure the consecutive change.** We compute the change in core neurons between two consecutive steps (i.e., between step $t-1$ and step $t$), denoted by $d_t$. This is essentially one minus the Jaccard overlap coefficient:
> > >  $d_t = 1 - \frac{|P_t \cap P_{t-1}|}{|P_t \cup P_{t-1}|}$
> > >
> > > 3. **Smooth the change sequence.** To more accurately measure the overall influence of newly added tokens on the core neurons, we smooth the sequence $d_t$ using an exponentially weighted moving average (EWMA), yielding the smoothed indicator $S_t$ (with $\lambda = 0.2$ in our experiments): $S_t = \lambda d_t + (1-\lambda)S_{t-1}$
> > >
> > > 4. **Automatically route the strategy based on the change magnitude.** The model then uses the current smoothed indicator $S_t$ to automatically determine the initialization mode:
> > >
> > >    - **Stability-guided:** If $S_t$ is small, this indicates that as the input grows longer, the core neurons have become stable and no longer change substantially.
> > >    - **Similarity-guided:** If $S_t$ is large, this indicates that the newly added input is still causing significant changes in the core neurons.
> > >
> > > ### Regarding the Clustering Details
> > >
> > > During inference, for a given input sample, we perform the following steps:
> > >
> > > 1. **Feature extraction and nearest-neighbor search.** We extract the sample's representation vector $z_x$ and identify the nearest cluster centroid $k^* $ :
> > >    $k^* = \arg\min_k |z_x - \mu_k|_2^2$
> > >
> > > 2. **OOD detection and initial routing.** We compare the distance to $k^*$ against a predefined threshold $\tau_{ood}$:
> > >
> > >    - **Distance** $< \tau_{ood}$ **(in-distribution):** Directly use the corresponding cluster-level core neuron set.
> > >    - **Distance** $\geq \tau_{ood}$ **(OOD):** Trigger a fallback, temporarily reverting to full-activation mode during decoding.
> > >
> > > 3. **Stable-state capture.** For OOD samples, full-activation decoding continues until reaching a stable state (indicated by the $S_t$ criterion). The system then records the stable core-neuron set $\hat C^l$ and representation vector $\hat z$.
> > >
> > > 4. **Incremental online memory update.** Using $\hat z$, we re-evaluate cluster distances to update the memory bank. We either create a new cluster (if $\hat z$ remains far from all clusters) or incrementally update the nearest cluster's centroid and neuron statistics (if within an acceptable range).
> > >
> > > In the revised paper, we will introduce these technical details more explicitly to ensure clarity. We again sincerely thank the reviewer for the time and effort devoted to evaluating this paper.

---

### Official Review · Reviewer_7UtX · 2026-03-12

**Soundness:** 3
**Presentation:** 2
**Significance:** 4
**Originality:** 3
**Overall Recommendation:** 3
**Confidence:** 4

**Summary:**

This work introduces the concept of "core neurons" in LLMs and constructs the SparseInfer to eliminate redundant neuron computations, thereby achieving the goal of improving computational efficiency.

**Compliance With Llm Reviewing Policy:**

Affirmed.

**Key Questions For Authors:**

See Weaknesses

**Limitations:**

1. This work does not discuss the consequences when the wrong core neurons are chosen.
2. See Weaknesses

**Strengths And Weaknesses:**

Strengths:
1. This work proposed an interesting concept of the "core neurons", which is sound and useful.
2. The results partially support authors' claim, which can influence future research or applications.

Weaknesses:
1. What is the definition of "stability", which is very important in this work. Does it mean that the core neurons are unchanged (neuron view), or the semantics are unchanged (sentence view)? Can we understand it this way: the meaning stays the same before and after expanding the sentence? If so, does that mean their core neurons are the same (stable)?
Also, how is the "similarity" defined? For semantically similar sentences, does similar core neuron activity imply that these neurons are spatially adjacent?
2. What is the shape of c_a^\beta(S_i)?
3. Should provide the t-SNE with the original neurons vs. c_a^\bata(S_i) in fig. 4
4. The legends in fig. 3(b)(c) are unclear. It seems that, as the number of tokens increases, both semantic similarity and core neuron similarity decrease. This appears to contradict Insight -1. This leads us back to the initial question: What defines "stability"? Does it refer to the stability of the sentences' semantics, or the stability of the core neurons themselves?

---

> ### Author Rebuttal · Authors · 2026-03-30
>
> We sincerely thank the reviewer for the time and effort spent reviewing our paper. **During the rebuttal period, we conducted many additional experiments to further strengthen our paper. Please refer to https://drive.google.com/file/d/1mzeIdgzXQ4tnFyT39dPNJ9yfYF_cMJdn/view?usp=sharing**
>
> ## 1. Definition of "stability" and "similarity"
>
> We apologize for the confusion caused. Overall, the reviewer's understanding is correct.
>
> **Core-neuron stability** means that when the sentence is extended, the sentence-level core neurons remain the same or change only slightly. In other words, if we continue adding tokens to a sentence, a stable input should keep a largely consistent set of core neurons before and after the extension. **Semantic stability** means that the meaning of the sentence remains consistent when the sentence is extended. In our experiments, we found that these two properties usually appear together: if the sentence meaning does not change much after extension, its core neurons usually do not change much either.
>
> In contrast, **Core-neuron similarity** measures the overlap between the sentence-level core neurons of *different* input samples. We found that it is strongly correlated with **Semantic similarity**: if two sentences have similar meanings, they usually have a high overlap in their core neurons, which suggests that their important neurons are close in activation space.
>
> Definitions are as follows.
>
> 1. **Core-neuron Stability:** The degree to which sentence-level core neurons remain unchanged as tokens are added.
> 2. **Semantic Stability:** The cosine similarity between S-BERT embeddings of a sentence before and after expansion. High semantic stability typically implies high core-neuron stability.
> 3. **Core-neuron Similarity:** The overlap between core neurons of two different samples.
> 4. **Semantic Similarity:** The S-BERT cosine similarity between different samples.
>
> ------
>
> ## 2. What is the shape of $C_{\alpha}^{\beta}(s_i)$?
>
> Thank you for pointing this out.
> By definition, $C_{\alpha}^{\beta}(s_i)$ is the set of selected neuron indices at layer $i$, i.e., a subset of ${1, \dots, N}$, where $N$ is the number of neurons in that layer.
> In implementation, we equivalently represent it as an $N$-dimensional binary mask with shape $[N]$. The $n$-th entry is 1 if and only if neuron $n \in C_{\alpha}^{\beta}(s_i)$, meaning that the $n$-th neuron is a core neuron. Otherwise, the entry is 0, meaning that this neuron is not selected as a core neuron .
>
> ------
>
> ## 3. Should provide the t-SNE of the original neurons vs. $C_{\alpha}^{\beta}(S_i)$ in Fig. 4
>
> We thank the reviewer for this helpful suggestion.
> In Fig. 4, each point represents a sample. We map its sentence-level core neurons to a high-dimensional one-hot vector for t-SNE. For the "original neurons" baseline, we record all neurons activated by *any* token in the sample, applying the same one-hot vectorization and t-SNE pipeline for comparison.
>
> **The results are shown in Fig. 4 of our additional experiments (https://drive.google.com/file/d/1mzeIdgzXQ4tnFyT39dPNJ9yfYF_cMJdn/view?usp=sharing).** We observe that the original neurons also show some semantic structure, in the sense that samples from different topics tend to form clusters. However, this pattern is less clear than that of the core neurons. This suggests that neuron activation is indeed related to semantics, but core neurons make this relation much more explicit.
>
> ------
> ##  4. The legends in Fig. 3(b)(c) are unclear.
>
> We thank the reviewer for this comment. What we intended to convey is not similarity must increase or remain unchanged as more tokens are added. Instead, our point is that **when the semantics of a sentence change less during continuous extension, its core neurons also remain more stable.** In this sense, semantic stability and core-neuron stability are positively correlated.
>
> More specifically, in Fig. 3(b)(c), the legend values length = 8/64/256 indicate the length of the **original** sentence, while the x-axis shows the number of **appended** tokens. Therefore, these figures should be interpreted as follows: when the original sentence is longer, its semantics are already more complete and better specified, so both semantic similarity and core-neuron similarity remain at a higher level as new tokens are appended. In contrast, when the original sentence is short, the appended content is more likely to change or enrich its meaning, and thus both similarities decrease more noticeably.
>
> At the same time, we cannot assume that a longer input will always have more stable core neurons. In actual inference, we monitor the activation behavior of the input sample  and dynamically decide whether to use the similarity-guided or stability-guided strategy. **More technical details about adaptive inference can be found in our response to Reviewer 6bKQ, Question 2.**
>
> **We will add  all these clarifications and experiments in the revised version.**

---

### Official Review · Reviewer_daGN · 2026-03-13

**Soundness:** 4
**Presentation:** 3
**Significance:** 2
**Originality:** 4
**Overall Recommendation:** 5
**Confidence:** 5

**Summary:**

This paper introduces SparseInfer, an MLP-free, sentence-level adaptive sparse activation framework designed to accelerate Large Language Model (LLM) inference. The authors point out that existing token-level sparse inference methods (e.g., PowerInfer, DejaVu) incur significant computational overhead from auxiliary MLP predictors and suffer from irregular memory accesses due to frequently changing activation maps.

To overcome these limitations, the paper proposes identifying a fixed set of "core neurons" for an entire sentence. The authors empirically demonstrate two key insights: (1) Stability: core neurons tend to stabilize as the input sentence lengthens and its context becomes clear; and (2) Similarity: sentences with similar semantics (topics) tend to activate similar subsets of core neurons.

Leveraging these insights, SparseInfer predicts the sentence-wise core neurons during the pre-filling stage using either a stability-guided or similarity-guided approach, depending on the input's characteristics. During the autoregressive decoding stage, these core neurons remain fixed, entirely eliminating the need for per-token prediction and dynamic memory loading. Extensive experiments demonstrate that SparseInfer maintains comparable task performance while achieving up to a 10.33x speedup over standard Huggingface implementations and outperforming existing MLP-based sparse methods on various benchmarks.

**Compliance With Llm Reviewing Policy:**

Affirmed.

**Final Justification:**

The authors have effectively addressed my concerns by introducing SparseInfer++ with concrete empirical evidence for OOD robustness.

**Key Questions For Authors:**

**1. Applicability and Future Ideas for Dynamic Batching Environments**

The current evaluation demonstrates impressive speedups and reduced memory footprints, presumably under a batch size of 1 (BS=1). However, in real-world LLM serving, dynamic batching groups multiple requests with diverse semantics. Since SparseInfer fixes a unique set of core neurons for each specific sentence, handling a batch of requests with completely different topics seems structurally challenging. Do you have any additional ideas, architectural extensions, or future plans to adapt SparseInfer for batched inference without losing the proposed sparsity benefits?

**2. Generalizability Limits of Static K-Means in Similarity-Guided Prediction**

For unstable or short inputs, the similarity-guided approach relies on K-Means clustering of the 25th layer's activations to determine the topic (Appendix A.4.3). However, these cluster centroids are computed offline based on specific calibration datasets. If the model receives a truly out-of-domain (OOD) or zero-shot prompt representing a completely novel topic that does not align with any pre-calculated centroid, how does the system ensure that an appropriate set of core neurons is selected? Could you provide empirical evidence or discussion on how SparseInfer performs on extreme OOD inputs where the static K-Means mapping might fail?

**Limitations:**

YES

**Strengths And Weaknesses:**

**1. Soundness**
* **Strengths:** The empirical observations regarding "core neurons" are highly insightful. The authors provide solid visual and quantitative evidence (e.g., t-SNE plots) showing that the active neuron subset stabilizes as context length increases, and that sentences with similar semantics cluster into similar core neuron sets. The shift from token-level to sentence-level sparsity is a logical and creative approach to resolve the MLP overhead and memory thrashing issues of prior works.


**2. Presentation**
* **Strengths:** The manuscript is visually intuitive, well-structured, and easy to follow. Figures 1, 3, and 4 effectively communicate the abstract concepts of core neuron stability and similarity to the reader.
* **Weaknesses:** The presentation of the method's limitations could be more transparent. A critical operational constraint—the potential breakdown of the method during unexpected semantic shifts—is relegated to Appendix A.2 (as "atypical" or "malicious" inputs). Moving this discussion to the main text would provide a more balanced understanding of the method's boundaries.

**3. Significance**
* **Strengths:** Accelerating LLM inference on resource-constrained, consumer-grade hardware is a highly relevant problem. For local environments or edge-device deployments, achieving substantial speedups without heavy re-training provides immense practical utility.
* **Weaknesses:** The scope of impact appears limited to specialized edge-case scenarios (i.e., isolated requests with constrained topics). Because the method assigns a unique, fixed subset of weights per sentence, it faces structural challenges in supporting high-throughput, general-purpose datacenter environments where handling diverse, batched requests and dynamic context shifts is essential.

**4. Originality**

The originality of this work is highly commendable. Shifting the paradigm from token-level dynamic sparsity (which relies on heavy MLP predictors) to a static, sentence-level "core neuron" approach is a highly creative synthesis of existing sparsity concepts. Uncovering the relationship between semantic similarity and neuron activation sets opens up novel perspectives for future research in model interpretability and efficient inference.

---

> ### Author Rebuttal · Authors · 2026-03-30
>
> We sincerely thank the reviewer for the time and effort spent reviewing our paper. **During the rebuttal period, we conducted many additional experiments to further strengthen our paper. Please refer to https://drive.google.com/file/d/1mzeIdgzXQ4tnFyT39dPNJ9yfYF_cMJdn/view?usp=sharing**
>
> ## 1. Method's limitations could be more transparent.
>
> We thank the reviewer for pointing this out. We agree that the limitations of our method should be discussed more clearly and more transparently.
>
> In fact, during the rebuttal period, we further extended SparseInfer to improve its robustness in real applications, and propose SparseInfer++. **More technical details can be found in our response to Reviewer 6bKQ, Question 2.** In general, we add three online adaptive mechanisms to SparseInfer:
>
> 1. Automatically choosing the core-neuron prediction strategy based on the stability of the input;
> 2. Adaptively updating core neurons during decoding to handle long-chain reasoning or semantic drift;
> 3. Using incremental K-Means to continuously update cluster centers and the number of clusters, while detecting and absorbing OOD inputs during inference.
>
> **In the revised paper, we will move both this applicability boundary and our solution into the main text**, so that readers can have a more complete and balanced understanding.
>
> ---
>
> ## 2. Applicability and Future Ideas for Dynamic Batching Environments
>
> We thank the reviewer for raising this important point. We agree that in a standard dynamic batching setting, as the batch size grows and the semantic diversity within the batch increases, the core-neuron sets of different samples tend to be merged through union operations. This naturally weakens activation sparsity. Therefore, for ordinary heterogeneous batches, SparseInfer is indeed not a naturally ideal solution.
>
> However, since SparseInfer determines the required neurons for each sentence during the prefilling stage, and since core neurons are closely related to semantics, we currently have two preliminary ideas for extending the method.
>
> **The first idea is to apply a lightweight semantic grouping step before dynamic batching**, so that samples with similar topics or meanings are placed into the same batch whenever possible. Since semantically similar samples often share more core neurons, this may greatly reduce the loss of sparsity caused by taking unions in heterogeneous batches.
>
> **The second idea is to first perform full activation for each sample during prefilling and determine its final neuron set**, and then reorganize these sparse activation patterns into a matrix form suitable for batch computation before performing batched sparse matrix multiplication. Of course, this direction would require further system and kernel design.
>
> These two ideas are still preliminary, and we plan to study them further in future work to enable efficient activation-sparse inference in batch settings. We sincerely thank the reviewer for this important question, which is very helpful for improving the practical deployment value of our method.
>
> ---
>
> ## 3. Generalizability Limits of Static K-Means in Similarity-Guided Prediction
>
> We thank the reviewer for this key question. During the rebuttal period, we further improved SparseInfer around this issue and developed SparseInfer++. One of its main improvements is a mechanism for handling OOD inputs and the limited generalization of static K-Means. **More technical details can be found in our response to Reviewer 6bKQ, Question 2**
>
> SparseInfer++ replace static K-Means with an online clustering mechanism. It can continuously monitor OOD inputs during inference and dynamically update both the cluster centers and the number of clusters.
>
> **Dynamic Mechanism:**
>
> - **Detection:** For a new input $z_x$, if its distance to the nearest centroid exceeds a threshold $\tau_{ood}$, it is flagged as OOD.
> - **Fallback:** OOD samples temporarily fall back to **full activation** to ensure accuracy.
> - **Update:** Once the OOD input’s core neurons stabilize, the system either creates a new cluster or incrementally updates an existing one, adaptively expanding its "clustering memory."
>
> **Validation (Fig. 3 of additional experiments):** We tested this by feeding Math and Coding data (OOD) into a model pre-trained only on AG News (4 categories). SparseInfer++ successfully identified the new distributions and formed stable new clusters during inference. This makes the system significantly more robust for real-world deployment where data distributions evolve. We will include these mechanisms in the revised manuscript.
>
> We thank the reviewer again for this suggestion. In the revised paper, we will include these mechanisms in SparseInfer++, so that readers can better understand the practical scope and behavior of the method.

---

> > ### Author Rebuttal · Reviewer_daGN · 2026-04-04
> >
> > The authors have effectively addressed my concerns by introducing SparseInfer++ with concrete empirical evidence for OOD robustness, so I am raising my score.

---

> > > ### Author Response · Authors · 2026-04-04
> > >
> > > Dear Reviewer,
> > >
> > > Thank you very much for your positive evaluation of our work and for raising your score. We are very glad to know that we have addressed your concerns.
> > >
> > > Your valuable suggestions have helped make this paper much more solid. We sincerely appreciate the time and effort you devoted to reviewing our paper.
> > >
> > > Best regards,
> > >
> > > The Authors

---

### Decision · Program_Chairs · 2026-04-30

**Decision:**

Accept (regular)

**Comment:**

The paper introduces SparseInfer, an MLP-free sentence-level adaptive sparse activation method that identifies a fixed set of “core neurons” per input sentence during pre-filling and reuses them throughout decoding to eliminate per-token overhead. The idea is elegant and shown to work. Reviewer recommendations were two Accepts, one Weak Accept, and one Weak Reject. As main weaknesses, reviewers mentioned limited model diversity (OPT/LLaMA). After providing the rebuttal, 3 reviewers marked their concerns “fully resolved” with scores raised. The reviewer with the lowest score did not participate in the post-rebuttal discussion. With this, I recommend acceptance of the paper to the ICML program, as efficient inference will be of high interest to the community.